# Long Short-Term Memory Networks for Enhancing Real-time Flood Forecasts: A Case Study for an Underperforming Hydrologic Model

Sebastian Gegenleithner[1,2,*], Manuel Pirker[1,*], Clemens Dorfmann[2], Roman Kern[3], and Josef Schneider[1]

[1]Graz University of Technology, Institute of Hydraulic Engineering and Water Resources Management, Stremayrgasse 10/II, 8010 Graz, Austria
[2]flow engineering, Lessingstraße 30, 8010 Graz, Austria
[3]Graz University of Technology, Institute of Interactive Systems and Data Science, Sandgasse 36/III, 8010 Graz, Austria
[*]These authors contributed equally to this work.
Correspondence: Sebastian Gegenleithner (s.gegenleithner@gmail.com) and Manuel Pirker (manuel.pirker@tugraz.at)

**Abstract.** Flood forecasting systems play a key role in mitigating socio-economic damages caused by flood events. The majority of these systems rely on process-based hydrologic models (PBHMs), which are used to predict future river runoff. Many operational flood forecasting systems additionally implement models aimed at enhancing the predictions of the PBHM, either by updating the PBHM's state variables in real-time or by enhancing its forecasts in a post-processing step. For the latter, especially AutoRegressive Integrated Moving Average (ARIMA) type models are frequently employed. Despite their high popularity in flood forecasting, studies have pointed out potential shortcomings of ARIMA-type models, such as a decline in forecast accuracy with increasing lead time. In this study, we investigate the potential of Long Short-Term Memory (LSTM) networks for enhancing the forecast accuracy of an underperforming PBHM and evaluate whether they are able to overcome some of the challenges presented by ARIMA models. To achieve this, we developed two hindcast-forecast LSTM models and compared their forecast accuracies to that of a more conventional ARIMA model. To ensure comparability, one LSTM was restricted to use the same data as ARIMA (eLSTM), namely observed and simulated runoff, while the other additionally incorporated meteorologic forcings (PBHM-HLSTM). Considering the PBHM's poor performance, we further evaluated if the PBHM-HLSTM was able to extract valuable information from the PBHM's results by analyzing the relative importance of each input feature. Contrary to ARIMA, the LSTMs were able to mostly sustain a high forecast accuracy for longer lead times. Furthermore, the PBHM-HLSTM also achieved a high prediction accuracy at flood event runoff, which was not the case for ARIMA and the eLSTM. Our results also revealed that the PBHM-HLSTM to some degree relied on the PBHM's results, despite its mostly poor performance. Our results suggest that LSTM models, especially when provided with meteorologic forcings, offer a promising alternative to frequently employed ARIMA models in operational flood forecasting systems.

## 1 Introduction

Floods are among the most common and most destructive natural disasters around the world (Yaghmaei et al., 2020). Alongside other mitigation measures, flood forecasting systems play a key role in increasing resilience to such events. In principle, flood forecasting systems enable the prediction of future river runoff, empowering decision-makers and emergency forces to

implement effective early countermeasures in the case of flooding events. Examples of such flood forecasting systems are given by Werner et al. (2009), Addor et al. (2011), Nester et al. (2016), Borsch et al. (2021), or Nearing et al. (2024).

To date, most operational flood forecasting systems are built around process-based hydrologic models (PBHM). These models predict future river runoff by utilizing conceptual or more physically based approaches that depict the individual components of the hydrologic cycle in the catchment. In recent years, many researchers have proposed solely data-driven models as an alternative to PBHMs. Particularly, models based on Long Short-Term Memory networks (LSTM, Hochreiter and Schmidhuber, 1997) have gained recognition for their capabilities in accurately modeling river runoff. For example, Kratzert

et al. (2019b) demonstrated that their LSTM model was able to outperform two PBHMs across multiple gauged but also ungauged catchments. Although data-driven models have proven to be a viable alternative to PBHMs for modeling river runoff, they are yet rarely applied as the core component in operational flood forecasting systems (Nevo et al., 2022).

The primary task of PBHMs employed in operational flood forecasting systems is predicting a sequence of future runoff values. The length of this sequence is chosen based on the characteristics of the catchment and is referred to as the forecast

horizon. For the chosen forecast horizon, the PBHM derives the runoff forecasts based on meteorologic quantities as well as its current system state at the beginning of the forecast horizon, e.g., the state of the snow cover, the soil moisture, or the available water below and above the surface (river runoff). A common practice in flood forecasting is to use real-time observations of these state variables, evaluate how the model was able to replicate them in the past, and use this knowledge to enhance the model's forecasts. Considering the available literature, the most relevant forecast-enhancing strategies can be grouped as

follows: (I) State updating: The basic idea behind this concept is to use observational data to update parts of the hydrologic model in real-time, allowing it to more accurately reflect the true state of the system. Commonly applied methods for state updating in flood forecasting include variants of the Kalman Filter (Kalman, 1960) or also Particle Filters, as demonstrated by Weerts and El Serafy (2006). (II) Error correction: These methods use observations of one or multiple state variables, mostly river runoff, to enhance the hydrologic model's forecasts in a post-processing step. Especially, models belonging to

the AutoRegressive Integrated Moving Average (ARIMA) family are frequently employed for this purpose. However, despite their high popularity, numerous studies have pointed out the potential limitations of these models in hydrologic modeling applications.

Firstly, ARIMA models often exhibit a decline in forecast accuracy with increasing lead time. For instance, Brath et al. (2002) demonstrated that the forecast accuracy of an adaptively updated ARIMA-type model degraded to match the accuracy

of the not-updated model after six time steps. A less significant performance decrease was observed for an ARIMA-type model that was calibrated with a split-sample strategy. Similarly, Broersen and Weerts (2005) demonstrated that their employed ARIMA-type models were able to significantly increase the prediction accuracy within the first day, while for further ahead predictions only slight differences were found to forecasts corrected with the mean runoff over the last three weeks. Secondly, ARIMA models struggle to provide accurate forecasts for flood event runoff when the underlying hydrologic model fails to

give an adequate initial estimation, as for example shown by Liu et al. (2015). In their study, Liu et al. (2015) assessed the predictive skills of an ARIMA-corrected PBHM for a total of four significant flood events. While their model demonstrated a high forecast accuracy for events that were already captured well by the hydrologic model, it failed in one instance where

this was not the case. Reasonable forecasts for this event could only be obtained in the consecutive forecast step, followed by a rapid decline in forecast accuracy.

Recently, researchers have explored the potential of neural networks, particularly Recurrent Neural Networks (RNN), to enhance the results obtained from PBHMs, and the outcomes have been remarkably successful. Although the focus of their study was on model diagnostics, Rozos et al. (2021) demonstrated that RNNs and LSTMs, trained on meteorologic data and the PBHM's output, have the potential to enhance the model accuracy of underperforming PBHMs. In a large-sample study, Konapala et al. (2020) tested various LSTM variants to enhance the prediction accuracy of a PBHM. They found that overall
their hybrid LSTM models that incorporated the results of the PBHM outperformed both the PBHM and in most instances also a standalone LSTM. They also found that the highest improvements were achieved for catchments where the PBHM was underperforming. A comparable study was also conducted by Frame et al. (2021). In their study, the authors showed that runoff predictions could be improved by LSTM models that incorporated the results of the PBHM. However, they also demonstrated that these models, in many instances, were outperformed by a standalone LSTM that did not incorporate information obtained
by the PBHM.

Given the promising findings of the aforementioned studies, we recognize the substantial potential of neural networks to enhance the forecast accuracy of underperforming PBHMs employed in operational flood forecasting systems. Especially in aspects where ARIMA correction methods previously demonstrated shortcomings, such as maintaining a high forecast accuracy for longer lead times, or accurately correcting poor flood event predictions, neural networks might yield more accurate
forecasts. To test this hypothesis, we developed two LSTM model variants, both implemented with a hindcast-forecast architecture, similar to the one presented by Gauch et al. (2021) or Nevo et al. (2022), and compared their forecast performances to that of a conventional ARIMA model. To ensure comparability, one LSTM variant, eLSTM, was restricted to use the same input data as ARIMA, specifically observed runoff and that obtained by the PBHM. The second variant, PBHM-HLSTM, was implemented with the same architecture as the eLSTM but additionally incorporated meteorologic forcings. It has to be
mentioned that our ARIMA model relied on forecasting residuals, while both LSTM variants directly predicted future runoff. For the PBHM-HLSTM exclusively, we also evaluated the contribution, or in other terms, the relative importance, of each input feature to assess the added value of the PBHM's predictions on the final model forecasts. To summarize, the main research questions addressed in this study can be stated as follows: (I) How does the overall forecast accuracy of the LSTM models compare to that of ARIMA, particularly for longer lead times? In this regard, it will be interesting to see whether the
non-linear LSTM outperforms the linear ARIMA model when using the same input data, and to what extent the LSTM can leverage additional meteorologic inputs. (II) Can the LSTM models achieve a higher forecast accuracy than ARIMA at flood event runoff? Notably, high forecast accuracy at flood event runoff is crucial in operational flood forecasting settings. (III) Is the PBHM-HLSTM able to extract valuable information from the underperforming PBHM's results?

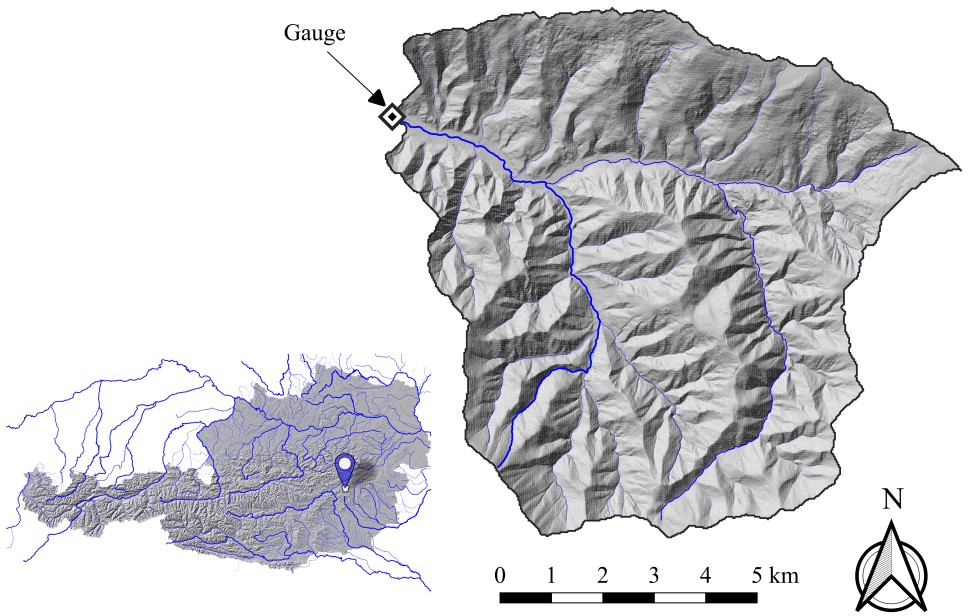

**Figure 1.** (Bottom left) Location of the study catchment in Austria. (Right) Outline of the study catchment (black line) including the gauging station (black and white diamond) and the main river network (blue lines). This figure was created using the following datasets: Umweltbundesamt GmbH (2022) and Land Kärnten (2019).

## 2 Study area and data

In this study, we investigated one medium-sized catchment located in the foothills of the Austrian Alps. The catchment drains an area of about $78\,\mathrm{km}^2$ and features elevations from approximately 600 to 1600 meters above sea level. The catchment features one gauging station operated by the Hydrographic Service of Styria (Austria). The mean annual runoff at the gauging station is approximately $1.0\,\mathrm{m}^3\mathrm{s}^{-1}$. The largest flood events in the catchment mostly occur during the summer months at a sub-daily time scale. Figure 1 provides an overview of the catchment's geographic location, its boundaries, the position of the gauging station, as well as the river network.

The catchment presented here was part of a broader study in which multiple catchments were modeled using a conceptual rainfall-runoff model (Gegenleithner et al., 2024a). Specifically, Gegenleithner et al. (2024a) employed the distributed wflow-hbv model (Schellekens, 2012). Due to the characteristics of the catchments investigated, the model was setup with a temporal resolution of 15 minutes. For most of the catchments presented in Gegenleithner et al. (2024a), the rainfall-runoff model displayed a high model accuracy with Nash-Sutcliffe Efficiencies (NSE) of 0.77 or higher and Kling-Gupta Efficiencies (KGE) of 0.83 or higher. However, for the catchment presented in this study, the model demonstrated a notably poorer performance. For the studied period (2011 - 2017) it merely achieved an NSE of 0.43, a KGE of 0.74, and a Percent Bias (PBIAS) of +16.0. For a detailed explanation of these performance metrics, refer to Appendix B. Additionally, the PBHM displayed significant

shortcomings in capturing the flood event runoff characteristics, i.e., the rising and falling limbs of the hydrographs as well as
the timing and magnitude of the maximum peak runoffs.

To develop our forecast models, we utilized the results of the PBHM at the gauge's location (see Fig. 1), denoted as $Q_{sim}$.
Additionally, we incorporated the observed discharge, henceforth referred to as $Q_{obs}$. For the PBHM-HLSTM exclusively, we
also included meteorologic forcings as an input. Specifically, 1x1 km rasters of total precipitation and near-surface temperature,
obtained from the Integrated Nowcasting through Comprehensive Analysis system (INCA, Haiden et al., 2011) and provided
by GeoSphere Austria, were utilized. From the raster data, we extracted the catchment's mean and maximum precipitation,
designated as $p_{mean}$ and $p_{max}$, along with its mean temperature, $T_{mean}$. Noteworthy, all datasets were available in 15-minute
intervals. A comprehensive overview of the used data and its key statistics is provided in Table A1.

## 3 Methodology

### 3.1 Development of the forecast models

For conducting this study, we developed a total of three model variants. The first model, ARIMA, relied on forecasting the
residuals between the simulated and observed runoff. Subsequently, the forecasted residuals were used to correct the PBHM's
forecasts. The second model, eLSTM, was based on a hindcast-forecast LSTM network, which similar to ARIMA used sim-
ulated and observed runoff to obtain the forecasts. However, contrary to ARIMA, the LSTM model directly predicted the
runoff in the forecast period. The third model, PBHM-HLSTM, was developed with the same architecture as the eLSTM, but
was supplied with additional meteorologic input, namely the mean and maximum catchment precipitation as well as its mean
temperature.

Considering the nature of the catchment investigated, all forecast models were developed with a temporal resolution of 15
minutes and a 24-hour forecast horizon, equivalent to 96 consecutive forecast steps.

### 3.1.1 Model optimization: Time series cross-validation

To optimize the hyperparameters of our ARIMA and LSTM models, we employed a blocked cross-validation strategy, as
recommended by Bergmeir and Benítez (2012). Furthermore, we chose an expanding window setup, which allowed us to
evaluate the model performances on a multitude of previously unseen data by progressively expanding the data available for
training, validation, and testing. Especially in hydrologic modeling applications, where the data exhibit considerable variability
(e.g., dry vs. wet years), this strategy can boost the model's performance on unseen data.

We implemented our cross-validation strategy by initially dividing the available time series into equally sized folds, i.e.,
subsets of the data. Each fold consisted of a sample size of $N = 34,903$, approximately equivalent to one year's worth of data.
This procedure resulted in seven folds corresponding to the years 2011 through 2017. Subsequently, we utilized these folds
to create a total of five cross-folds used for model training, validation, and testing. Following the expanding window strategy,

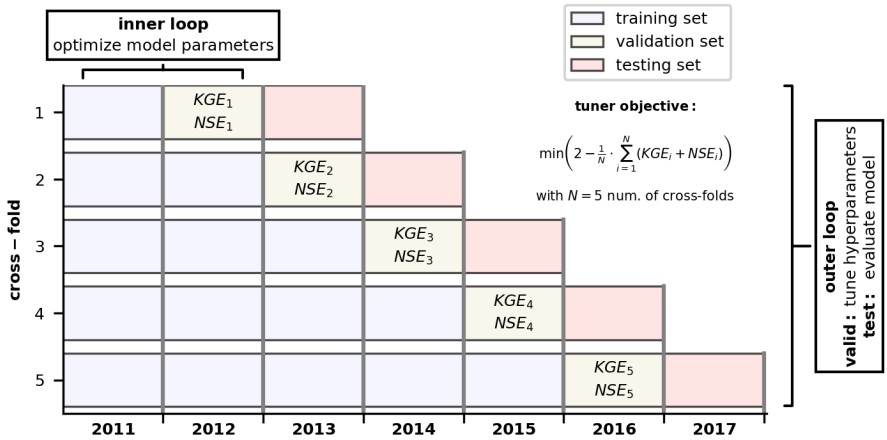

**Figure 2.** Blocked cross-validation strategy with expanding window setup. The parameters of the models were fitted within the inner loop while the hyperparameters were tuned in the outer loop, utilizing the validation fold of each of the five cross-folds.

each cross-fold was extended by one fold compared to the previous one. Within each cross-fold, the last and second-to-last
folds served as the testing and validation sets, while all preceding folds were used for model training.

For optimizing the models, we employed two loops. In the inner loop, the parameters of each model were optimized using the training and validation sets of each cross-fold. Following the recommendations of Tashman (2000), the models underwent retraining for each cross-fold. For the LSTM models, the hyperparameters were tuned in the outer loop. Thereby the performance of multiple candidate models was evaluated for the test sets, and the one that minimized the tuner objective function
was chosen for final deployment. For the objective function, we selected a combination of the NSE and KGE metrics. For a detailed description of the employed objective function, refer to Appendix C. For ARIMA the hyperparameters were defined by evaluating the PBHM's residuals, the overall model performance as well as ARIMA's model residuals. However, similar to the LSTM models, ARIMA's model parameters were fitted on the training sets of the cross-folds. A visual representation of the here presented methodology is provided in Fig. 2.

**3.1.2    AutoRegressive Integrated Moving Average model (ARIMA)**

ARIMA-type models are widely used for predicting hydrometeorologic time series such as precipitation or runoff (Brath et al., 2002; Broersen and Weerts, 2005; Liu et al., 2015; Khazaeiathar et al., 2022). ARIMA models are commonly denoted as $ARIMA(p, d, q)$, where $p$ is the order of the autoregressive part, $d$ is the differentiation order, and $q$ represents the order of the moving average component. In other words, the values of $p$ and $q$ indicate the number of previous values considered for making
the forecasts, and $d$ specifies the number of differentiation operations applied to the original time series. The here presented ARIMA model relies on forecasting the residuals of the PBHM's simulated runoff and that observed at the gauging station,

i.e., $e = Q_{sim} - Q_{obs}$. The forecasted residuals, $\hat{e}$, are then used to correct the PBHM's forecasts. A visual representation of this procedure is given in Fig. 3.

The ARIMA model presented in this study was developed by using the Python Statsmodels library (Seabold and Perktold, 2010). We assumed the PBHM's residuals to be approximately Gaussian. Furthermore, we assumed that the PBHM's residuals are correlated, stationary (or can be made stationary by ARIMA), and preferably close to homoscedastic. On closer inspection, we found that the residuals exhibited a high degree of heteroscedasticity, which could be stabilized by applying a Box-Cox transformation (Box and Cox, 1964) to the PBHM's results and the observed runoff prior to computing the residuals, as for example shown by Li et al. (2021). A fixed $\lambda$-value of 0.2 was used for the Box-Cox transformation, which has proven itself in hydrologic model applications (e.g., Li et al., 2021; Engeland et al., 2010). Noteworthy, the Box-Cox transformation also improved Gaussianity. For a detailed statistical evaluation of the residuals, refer to Appendix. A2. Stationarity was checked by investigating the Autocorrelation Function (ACF), which showed a slow decay over many lags, typically indicating some degree of non-stationarity (see Fig. A2, left). To make the time series stationary, we added one differentiation operation to the ARIMA model ($d = 1$), which was found to be sufficient for the data used in this study. Additionally to the ACF, we also computed the Partial Autocorrelation Function (PACF, see Fig. A2, right). The ACF and PACF were then used to get a first estimate of the $q$ and $p$ orders of the ARIMA model. Considering the narrow 5 % significance bounds and the rather low correlations, we iteratively determined the optimum model orders by evaluating ARIMA's overall model performance in the testing folds, whilst not overfitting the model. Additionally, we evaluated ARIMA's model residuals, which ideally should be independent, homoscedastic, and normally distributed. First, ARIMA's model residuals displayed some remaining correlation structures. Second, we also found that the residuals displayed some degree of non-Gaussianity and to a lesser degree heteroscedasticity, independent of the model configuration used. To summarize, the optimum model configuration for the ARIMA model presented in this study was $ARIMA(5, 1, 6)$.

Contrary to other studies (e.g., Broersen and Weerts, 2005), our ARIMA model was not retrained adaptively, i.e., in each forecast step. Instead, ARIMA's model coefficients were determined by utilizing the entire training time series of each cross-fold (see Sect. 3.1.1) and the resulting coefficients were used for the forecasts in the validation and test sets. Notably, such an approach was also employed by Brath et al. (2002).

### 3.1.3  Hindcast-forecast Long Short-Term Memory network (PBHM-HLSTM & eLSTM)

Long Short-Term Memory Networks (Hochreiter and Schmidhuber, 1997) are a special form of Recurrent Neural Networks (RNNs). They are specifically designed to address the common issue of vanishing gradients that are often encountered during the training process of RNNs. RNNs process sequential data by maintaining hidden states $H$ that retain information from previous inputs, allowing them to capture temporal dependencies. In addition, LSTMs possess cell states $C$ and incorporate three gates - namely, the input gate for controlling incoming information to the cell state, the output gate for regulating information passage to the hidden state, and the forget gate for determining the retention or clearance of stored information in the cell state.

The LSTM models presented in this study were developed using TensorFlow (Abadi et al., 2015) and the Keras framework (Chollet et al., 2015). Both LSTM variants were implemented with a hindcast-forecast architecture, similar to the one presented

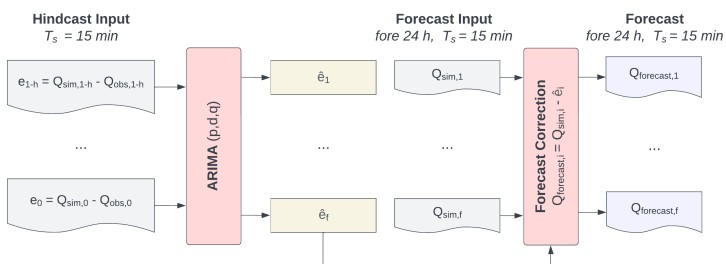

**Figure 3.** ARIMA architecture. The optimized $ARIMA(p, d, q)$ model utilized the residuals between the PBHM's results $Q_{sim}$ and the observed runoff $Q_{obs}$ in the past, $e$, to forecast the residuals in the forecast period $\hat{e}$. Consequently, the forecasted residuals were used to correct $Q_{sim}$ in the forecast period. Noteworthy, $h$ and $f$ refer to the hindcast and forecast periods, respectively.

by Gauch et al. (2021) and Nevo et al. (2022). This architecture involved coupling two distinct LSTM layers, one for the hindcast period and one for the forecast period, respectively. The sequence-to-one hindcast LSTM learned patterns in the data of the past. Subsequently, the hindcast LSTM's last hidden $H_0$ and cell states $C_0$ were extracted and handed to a fully connected layer. The output of this layer was then used to initialize the first hidden $H_1$ and cell states $C_1$ of the sequence-to-sequence forecast LSTM. Besides information on the hindcast period, that was given by the states of the hindcast LSTM, the forecast LSTM included additional features available in the forecast period. The sequential output of the forecast LSTM was then flattened and passed through another fully connected layer to obtain the runoff forecasts for the next 24 hours. For this layer, we used the Rectified Linear Unit (ReLU) as the activation function, which prevented negative runoff forecasts.

To prevent data leakage, the models' input features were normalized based on statistics calculated from the first available year (2011). For the normalization, we used min-max scaling for the runoff and precipitation data, while z-score standardization was used for the temperature. The models were trained using the mean squared error (MSE) as a loss function. The hyperparameter tuning was conducted by employing the Adam optimizer (Kingma and Ba, 2017), which minimized a combined objective function consisting of the KGE and NSE metrics (see Appendix C).

The architecture presented in Fig. 4 was used to develop two model variants. The first variant, eLSTM, solely included the observed runoff in the hindcast as well as the simulated runoff in both the hindcast and forecast periods. The second model variant, PBHM-HLSTM, additionally included meteorologic forcings. Specifically, the catchment's mean and maximum precipitation as well as its mean temperature in both the hindcast and forecast periods were used. Additionally, PBHM-HLSTM incorporated runoff observations in the hindcast period and the PBHM's results in both the hindcast and forecast periods, respectively.

To optimize the models' hyperparameters, we employed a random grid search tuner (O'Malley et al., 2019) as the outer loop of the cross-validation strategy presented in Sect. 3.1.1. Auxiliary information on the parameters subjected to optimization as well as the models' final hyperparameters can be found in Appendix C.

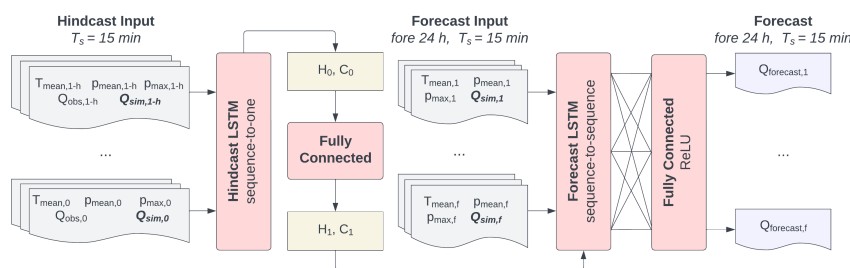

**Figure 4.** LSTM architecture. The optimized LSTM models incorporated the PBHM's simulations $Q_{sim}$ in both the hindcast and forecast periods as well as the observed runoff at the gauging station $Q_{obs}$. The PBHM-HLSTM exclusively incorporated the meteorologic quantities $p_{mean}$, $p_{max}$, and $T_{mean}$ in both the hindcast and forecast periods. The hidden and cell states of the hindcast LSTM ($H_0$ and $C_0$) were used to initialize the hidden and cell states of the forecast LSTM ($H_1$ and $C_1$). Noteworthy, $h$ and $f$ refer to the hindcast and forecast periods, respectively.

### 3.1.4 Sensitivity Analysis of Neural Networks - Integrated Gradients

To assess the importance of each input feature processed by the LSTM model, we used the Integrated Gradient Method (IG, Sundararajan et al., 2017). Noteworthy, this evaluation was exclusively conducted for the PBHM-HLSTM, which included all input features used in this study. The integrated gradients were evaluated for the model's output, which can be written as follows:

$$IG_i^{approx}(\boldsymbol{x}) = (x_i - x_i') \times \sum_{k=1}^{m} \frac{\partial F\left(\boldsymbol{x'} + \frac{k}{m}(\boldsymbol{x} - \boldsymbol{x'})\right)}{\partial x_i} \times \frac{1}{m} \tag{1}$$

where $\boldsymbol{x}$ is the input of interest, $F$ is the model, $\boldsymbol{x'}$ is the baseline (in our case a sequence of zeros as suggested by Kratzert et al., 2019a), $x_i$ is the input in the $i^{th}$ dimension, i.e., at the $i^{th}$ input node, and $m$ is the step size of the approximation of the integral (here 50 as suggested by Sundararajan et al., 2017). In our case, the output of the model is a sequence of size 96, representing the forecast steps. The number of input dimensions, i.e., input nodes, accumulates from five hindcast features (each with a sequence of size 48), and four forecast features (each with a sequence of size 96), resulting in 624 integrated gradients per output node and sample.

### 3.2 Model performance evaluation

We utilized the five cross-folds (2013 through 2017) presented in Sect. 3.1.1, specifically the test sets, for evaluating the performances of our forecast models. In alignment with the research questions addressed in this study, we conducted the following evaluations:

- How does the overall forecast accuracy of the LSTM models compare to that of ARIMA, particularly for longer lead times? To answer this question, we first evaluated each model's (ARIMA, eLSTM, and PBHM-HLSTM) annual performance, i.e., the overall performance for each of the five previously unseen testing years. For this evaluation, we utilized well-established metrics in hydrology, namely the NSE, the KGE, and the PBIAS. Additionally, we included the FHV high flow bias, which evaluates the model bias for the highest 2 % of the flow duration curve. The formulation of the FHV can be found in Appendix B alongside the other performance metrics. For each metric, we computed the annual average across the 24-hour forecast horizon as well as the individual values corresponding to the 96 forecast steps. Besides that, we also monitored the propagation of the mean absolute error (MAE) and the variability of the absolute errors (AE) for each lead time step. The variability was assessed by computing the standard deviation of the absolute errors for each forecast step. In general, models with a high forecast accuracy are expected to display an MAE close to zero and a low standard deviation.

- Can the LSTM models achieve a higher forecast accuracy than ARIMA at flood event runoff? This question was addressed by conducting a detailed investigation of each model's performance for the two largest flood events in each year. Specifically, we evaluated how well the models were able to capture the maximum peak runoff in both timing and magnitude. For this purpose, we computed the median peak magnitude error as well as the median temporal offset across all forecasts in a predefined evaluation window. To add to this, we also evaluated the distribution of the MAE and the variability of the absolute errors. This was done analogously to the methodology presented in the previous point, but for the highest 2 % of the runoff only.

- Is the PBHM-HLSTM able to extract valuable information from the underperforming PBHM's results? This question was addressed by evaluating the importance of each input feature by employing the IG method presented in Sect. 3.1.4. In accordance with the previous research questions, we evaluated the PBHM-HLSTM's overall feature importance as well as the importance at flood event runoff, again for the two largest flood events per year. The overall importance was assessed by calculating the integrated gradients from the sum of all values at the output nodes and was evaluated for all testing folds. On the other hand, the feature importance at flood event runoff was determined by computing the IG from the maximum value at the output nodes, which was evaluated for all samples when the maximum peak was present in the forecast horizon. This approach enabled us to assess the importance of each feature at different distances from the maximum runoff peak. For instance, how important is the observed runoff when the maximum peak is three steps away from the forecast origin, $t_0$.

## 4 Results

### 4.1 Overall model performance

#### 4.1.1 Annual average model performance

Evaluating the average annual model performances for the NSE, KGE, PBIAS, and FHV metrics showed that all model variants improved upon the underperforming PBHM's results. Each model's annual performance metrics, averaged over the 24-hour forecast horizon, are reported in Table 1.

The results revealed that the LSTM-based models excelled in terms of NSE and KGE, which was found to be especially true for the PBHM-HLSTM. For instance, the PBHM-HLSTM was able to achieve an average NSE value of 0.92 in 2013
compared to the 0.19 of the original PBHM. Even in the worst-performing year, 2017, the PBHM-HLSTM was able to elevate the average KGE and NSE values of the PBHM from 0.19 and -4.24 to 0.83 and 0.70, respectively. Overall, the PBHM-HLSTM was found to outperform both ARIMA and the eLSTM in terms of NSE and KGE in most of the years evaluated, and in the years where this was not the case, the differences in performance were marginal. A different image is drawn when investigating the models' bias metrics (PBIAS and FHV). Particularly in terms of PBIAS, ARIMA's performance was found to
be outstanding when compared to the other model variants. We found that this was due to ARIMA's exceptional performance in cases where the forecasts followed a clear pattern or trend, which in hydrologic model applications is often the case in baseflow conditions. Although ARIMA also showed a comparably high performance for the FHV bias, the performance gap to the LSTM models was less distinct. In fact, the PBHM-HLSTM showed the most consistent results in this regard, producing no significant outliers.

The significant performance gap between the PBIAS and the NSE and KGE metrics, however, suggested shortcomings in the forecasts obtained by ARIMA. The most straightforward way to identify these shortcomings was by dissecting the individual components of the KGE efficiency metric. This metric consists of three components that measure the linear correlation, the bias, and the variability between the simulated and observed runoffs. As expected, the KGE's bias term for the ARIMA forecasts was close to perfect. Also, the variability term did not signal systematic shortcomings compared to the results of the LSTMs.
However, regarding the linear correlation term, we found that the LSTM forecasts significantly outperformed those of ARIMA. According to Gupta et al. (2009), this term is majorly influenced by the model's ability to capture the peak timing as well as the rising and falling limbs of the hydrographs.

#### 4.1.2 Average model performance over lead time

Each model's performance was also assessed by monitoring the development of the NSE, KGE, PBIAS, and FHV metrics
across the 24-hours forecast horizon (96 consecutive time steps). The results of each testing year and metric are presented in Fig. 5.

As anticipated, both ARIMA's and the LSTMs forecasts surpassed the PBHM's results across most evaluated metrics and years. ARIMA, in particular, demonstrated an exceptional performance for both bias metrics, PBIAS and FHV. The only

**Table 1.** Average annual model performance comparison. Shown are the efficiency metrics KGE, NSE, PBIAS, and FHV. All metrics are averaged over the entire forecast horizon and are reported for each testing year. The best values per metric and year are highlighted in **bold**.

| year | PBHM | | | | ARIMA | | | | eLSTM | | | | PBHM-HLSTM | | | |
|------|------|-----|-------|------|------|-----|-------|------|------|-----|-------|------|------|-----|-------|------|
| | KGE | NSE | PBIAS | FHV | KGE | NSE | PBIAS | FHV | KGE | NSE | PBIAS | FHV | KGE | NSE | PBIAS | FHV |
| 2013 | 0.63 | 0.19 | +15.8 | +32.4 | **0.90** | 0.79 | **+0.4** | **-0.7** | 0.89 | 0.85 | -1.1 | +2.3 | 0.87 | **0.92** | -4.7 | +1.5 |
| 2014 | 0.74 | 0.49 | +15.6 | +6.9 | 0.89 | 0.79 | **+0.7** | **+5.8** | 0.85 | 0.90 | +2.1 | -13.4 | **0.94** | **0.95** | -3.1 | -8.8 |
| 2015 | 0.51 | 0.24 | +6.2 | +35.5 | 0.82 | 0.70 | **+0.6** | +22.4 | **0.88** | **0.90** | +1.3 | -5.4 | 0.87 | 0.88 | +7.7 | **+4.7** |
| 2016 | 0.74 | 0.51 | +10.5 | **-5.2** | 0.84 | 0.70 | **+0.3** | **-5.2** | 0.68 | 0.68 | -8.6 | -38.6 | **0.89** | **0.88** | -2.6 | -9.8 |
| 2017 | 0.19 | -4.24 | +60.0 | +74.5 | 0.60 | 0.22 | **+0.5** | +7.4 | 0.79 | 0.64 | -0.7 | **-1.0** | **0.83** | **0.70** | +12.2 | +10.1 |

exception was found to be ARIMA's high FHV in 2015. Also in terms of NSE and KGE, ARIMA showed outstanding forecast accuracy for the first couple of forecast steps. However, this initial accuracy showed to decline quickly with increasing lead time. This became particularly evident in 2017 when ARIMA's initial KGE dropped from 0.98 in the first prediction step to 0.60 in the last. An even more significant performance decrease was observed for the NSE metric, for which ARIMA achieved an initial value of 0.97 in the first step but 0.29 in the last. Compared to ARIMA, the LSTM models displayed a different forecast behavior. First, the bias metrics of both LSTM models were mostly higher when compared to those obtained by ARIMA, particularly the PBIAS. Interestingly, when solely judged by their bias metrics, both LSTM variants suggested more or less equal model performance, outperforming each other in some of the years used for evaluation. Arguably, the PBHM-HLSTM achieved more consistent forecasts considering that the eLSTM produced a significant FHV bias in 2016. Second, the LSTMs consistently performed worse than ARIMA in the first forecast steps, as suggested by both the KGE and NSE metrics. However, contrary to ARIMA, they tended to mostly sustain their initial accuracy across the forecast horizon. This was found to be most pronounced for the NSE, though it was also observed to a lesser degree for the KGE. For instance, even in the worst-performing year, 2017, the PBHM-HLSTM was able to uphold an NSE of 0.62 and a KGE of 0.73 across the 24-hour forecast horizon. Comparing the eLSTM and PBHM-HLSTM model variants, the latter clearly showed superior model performance when judged by the NSE and KGE metrics. Besides a few exceptions where both models performed on par, the PBHM-HLSTM outperformed the eLSTM across all years used for evaluation in this regard. This clearly highlights the added benefit of adding meteorologic forcings into model development.

In addition to the presented metrics used for model evaluation, we also measured the forecast performance by means of the mean absolute error (MAE) and the standard deviation of the absolute errors ($\sigma(AE)$). Both measures were evaluated across the forecast horizon and are shown in Fig. 6. Analogously to the results presented in Fig. 5, the trend of the mean absolute error (Fig. 6, left) displays ARIMA's high forecast accuracy in the first couple of prediction steps. However, it also reaffirmed its gradual decline in accuracy. Contrary to that, the LSTM model variants displayed larger errors in the first steps, but their decline in forecast accuracy was less pronounced. Interestingly, in some years (i.e., 2013, 2015, and 2017) the MAE of the LSTM-based models was found to be higher throughout the entire forecast horizon. At first glance, this contradicts the results presented in Fig. 5, particularly when focusing on the NSE metric. This apparent contradiction, however, can be explained

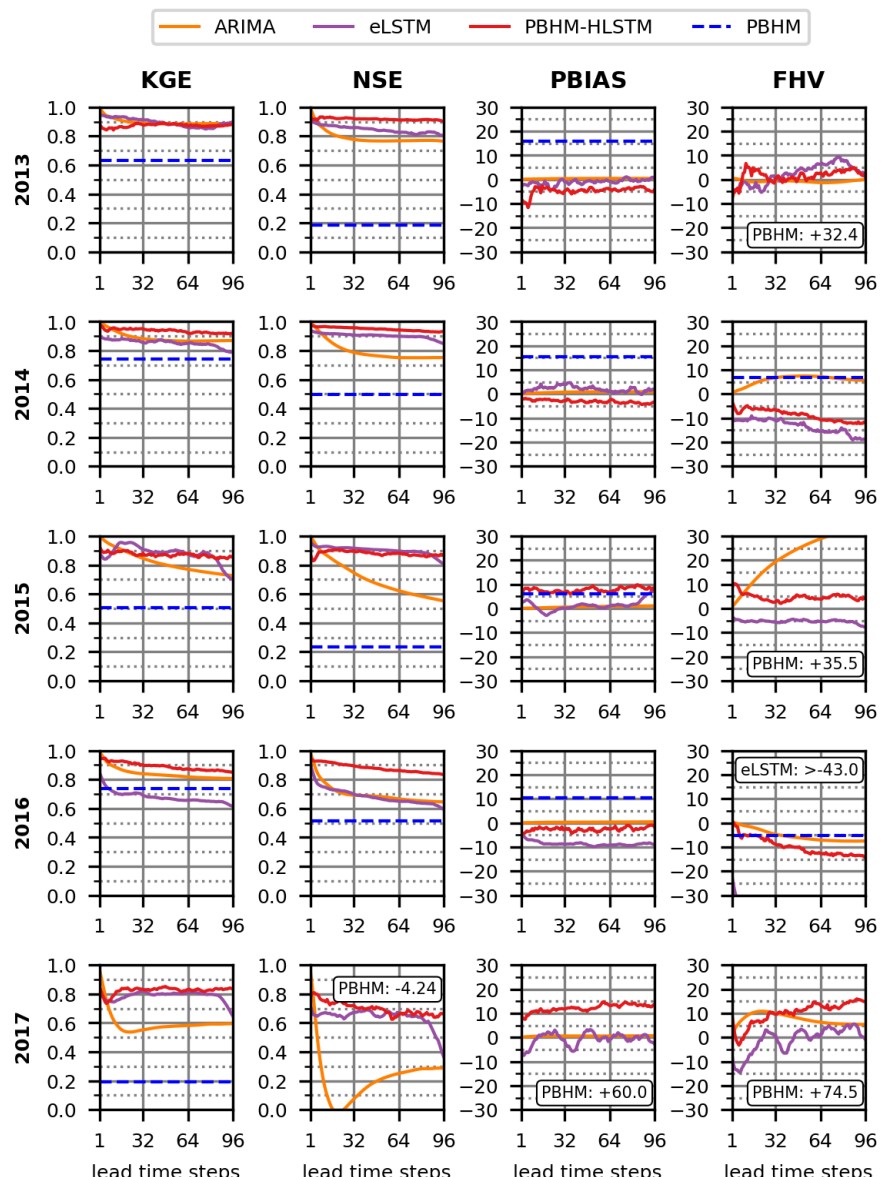

**Figure 5.** Development of the KGE, NSE, PBIAS and FHV metrics over the 24-hour (96 lead time steps) forecast horizon. Included are all developed model variants and all testing years.

by the variability of the errors shown in Fig. 6 (right). Unlike the LSTM-based models, the forecasts generated by ARIMA demonstrated significant variability in their errors. This indicates that while ARIMA produced highly accurate forecasts in some cases, it often yielded predictions that deviated substantially from the actual outcomes, especially for further ahead forecasts. In

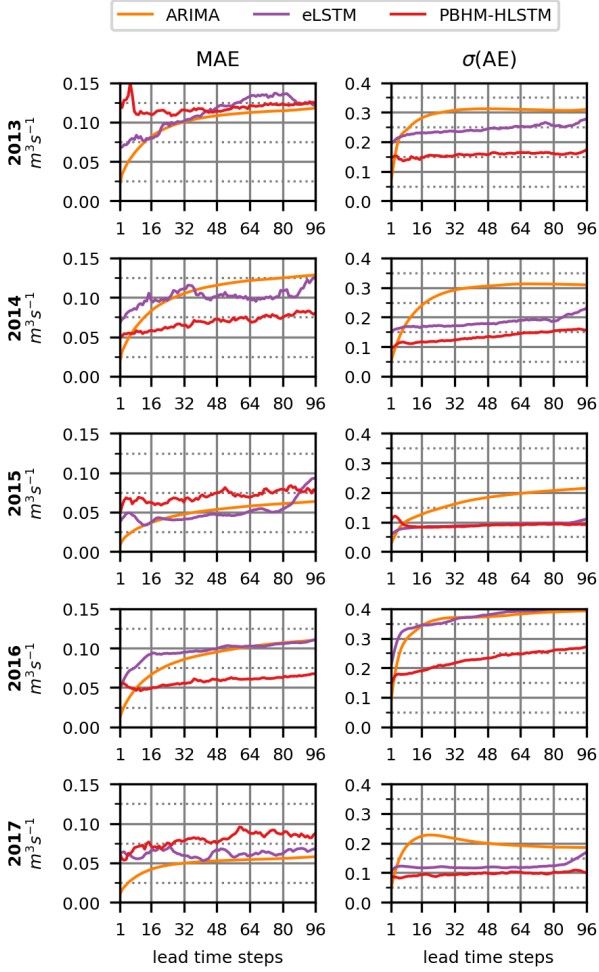

**Figure 6.** Development of the absolute errors for all flows. Shown are the MAE and the standard deviation $\sigma$ of the AE per testing year for the 24-hour forecast horizon (96 time steps).

contrast, both LSTM variants achieved a considerably lower error variance. Quantitatively, the LSTM-based models provided more reliable forecasts on average after three forecast steps (i.e., 45 minutes).

## 4.2 Performance for elevated river runoff

### 4.2.1 Peak timing and magnitude errors

For assessing the performances of our forecast models at flood event runoff, we determined the models' median peak magnitude and timing errors for the two largest flood events in each year. The magnitude error, $e_{peak}$, quantifies the median offset between

the maximum observed and simulated peak runoff across the evaluation window in percent. Similarly, the timing error $\Delta t$ measures the median temporal offset between the maximum observed and simulated peak runoff in number of time steps. Positive magnitude errors indicate model overestimation, while negative values suggest an underestimation. As for the timing errors, negative values indicate that the model predicted the maximum peak runoff earlier than observed, and positive values indicate the opposite. The results of this evaluation are presented in Table 2.

Upon initial inspection, the evaluation of the peak magnitude and timing errors reaffirmed the deficiencies of the PBHM in capturing the flood runoff dynamics. Especially, the substantial timing errors suggest shortcomings of the model in adequately depicting the characteristics of the flood event hydrographs. In terms of magnitude error, the PBHM achieved a median value of -49.9 %, predominately underestimating the observed peak runoff. Arguably, none of the investigated model variants was able to precisely pinpoint the magnitude of the flood events. However, by far the best performance in this regard was shown by PBHM-HLSTM, which achieved a median magnitude error of -27.5 %. Interestingly, the worst performance in this regard was shown by the eLSTM and ARIMA's results lying in between. It has to be mentioned that ARIMA, in contrast to eLSTM, generally exhibited a lower magnitude error in the first steps, which improved the overall value reported.

In terms of timing error, the ARIMA-corrected forecasts showed no improvement compared to the PBHM's original forecasts. In contrast, the eLSTM was able to majorly reduce the timing errors in the forecasts. Considering the fact that both models were supplied with the same input data clearly shows that the linear correction model (ARIMA) was not able to adequately transform the shape of the poorly depicted hydrographs of the PBHM. Comparing the timing errors of the eLSTM and PBHM-HLSTM reaffirmed the superiority of the PBHM-HLSTM. Specifically, the PBHM-HLSTM displayed a median timing error of merely two time steps across all events, which corresponds to 30 minutes in this study. In contrast, the median timing error of the eLSTM was found to be 5 time steps. Auxiliary information on the models' predictions for the largest flood events in each year can be found in Appendix. D.

### 4.2.2 Performance over lead time for elevated runoff

Analogously to the results presented in Fig. 6, we evaluated the development of the mean absolute error (MAE) and the standard deviation of the absolute errors ($\sigma(AE)$) across the forecast horizon, but only evaluated for the largest 2 % of the runoff. The results of this investigation are shown in Fig. 7.

While ARIMA often outperformed or matched the MAE of the LSTMs when considering all flows (see Fig. 6, left), the evaluation of the largest runoff values clearly demonstrates the superiority of the LSTM-based models (see Fig. 7, left). Particularly the PBHM-HLSTM, but to a lesser degree also the eLSTM, was able to achieve a considerably lower MAE compared to ARIMA. A similar picture was drawn by the variance of the absolute errors (see Fig. 7, right) for which the PBHM-HLSTM displayed considerably lower values than both ARIMA and the eLSTM.

**Table 2.** Comparison of the median peak magnitude error $e_{peak}$ (in percent) and timing error $\Delta t$ (in number of time steps) for the two largest flood events in each year. The smallest errors and offsets per event are highlighted in **bold**.

| year | event | obs. peak runoff $(\mathrm{m^3 s^{-1}})$ | PBHM $e_{peak}$ (%) | $\Delta t$ | ARIMA $e_{peak}$ (%) | $\Delta t$ | eLSTM $e_{peak}$ (%) | $\Delta t$ | PBHM-HLSTM $e_{peak}$ (%) | $\Delta t$ |
|------|-------|------|------|------|------|------|------|------|------|------|
| 2013 | 1st | 15.00 | -90.3 | 31 | -71.7 | 47 | -78.0 | 59 | **-38.4** | **2** |
|      | 2nd | 10.02 | **+13.3** | 20 | -26.1 | 19 | -40.0 | **-1** | -27.5 | -5 |
| 2014 | 1st | 7.27 | **+3.5** | 18 | +4.5 | 19 | -27.0 | **0** | -27.8 | -2 |
|      | 2nd | 6.23 | +23.3 | 16 | **+19.1** | 16 | -24.8 | **2** | -26.5 | 4 |
| 2015 | 1st | 5.85 | -62.5 | 36 | -29.9 | 36 | -66.7 | 10 | **-25.2** | **2** |
|      | 2nd | 3.33 | **+4.7** | 49 | +17.4 | 44 | -32.7 | **0** | -13.4 | 3 |
| 2016 | 1st | 17.94 | -73.6 | 26 | **-45.9** | 29 | -77.1 | 5 | -68.8 | **3** |
|      | 2nd | 9.99 | -45.4 | **18** | -56.7 | 20 | -65.5 | 27 | **-43.9** | 68 |
| 2017 | 1st | 9.21 | -49.9 | 25 | -48.9 | 48 | -54.5 | 3 | **-17.3** | **-1** |
|      | 2nd | 7.37 | -63.1 | 28 | -32.6 | 31 | -59.9 | 6 | **+8.4** | **1** |
| all folds | 1st |  | -62.5 | 26 | -45.9 | 36 | -66.7 | 5 | **-27.8** | **2** |
|      | 2nd |  | **+4.7** | 20 | -26.1 | 20 | -40.0 | **2** | -26.5 | 3 |
|      | both |  | -49.9 | 26 | -32.6 | 31 | -59.9 | 5 | **-27.5** | **2** |

## 4.3 Sensitivity analysis of the PBHM-HLSTM

### 4.3.1 Overall sensitivity

The average importance of each of the PBHM-HLSTM's input features for both the hindcast and forecast LSTMs is shown in Fig. 8. As anticipated, the PBHM-HLSTM heavily relied on past runoff observations $O_{obs}$ for deriving its forecasts. Interestingly, the importance of the observations seemed to decay exponentially with increasing distance to the forecast origin, $t_0$. As shown in Fig. 8, the influence of the observations almost dampened out after approximately 48 time steps. This means that the model gave increasingly more weight to observations close to $t_0$. Furthermore, in the annual average, the model seemed to

rely very little on past and future precipitation, $p_{max}$ and $p_{mean}$, most likely because both variables were zero or close to zero throughout most of the year. In comparison to that, the mean temperature $T_{mean}$ in both the hindcast and forecast had some influence on the predictions, most likely adding seasonality context to the model. The PBHM's simulated runoff was found to have the second highest impact on the forecasts, right after the observations. Particularly in the forecast period, the simulated runoff influenced the final predictions considerably.

Table 3 summarizes the normalized feature importances for all evaluation years, averaged over the hindcast and forecast periods, respectively. This evaluation reaffirmed that the model highly valued the observed runoff, which was found to be the most important feature for all years. Also consistent across all years was the high importance of the PBHM's simulated runoff,

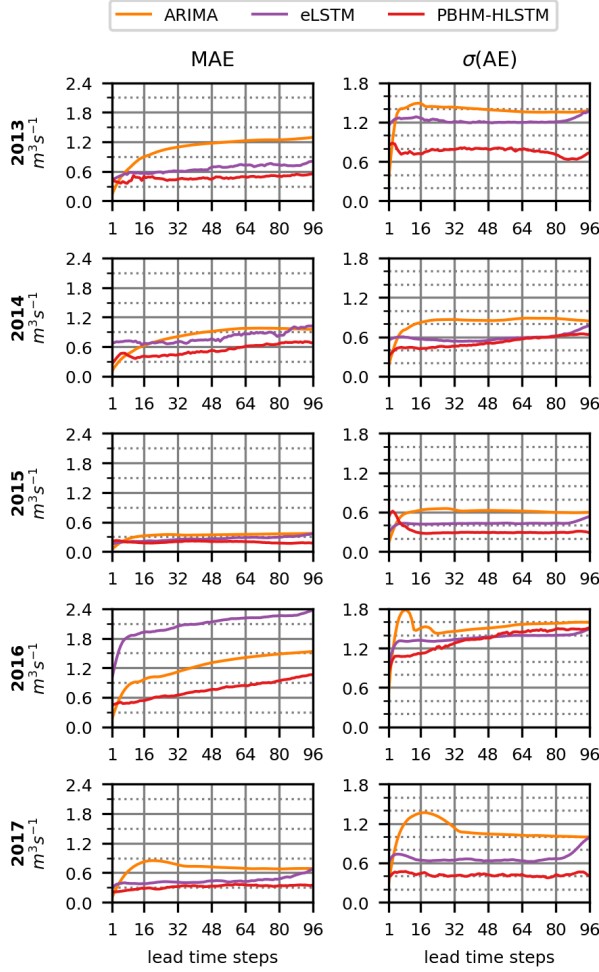

**Figure 7.** Development of the absolute errors for the largest 2 % of the runoff. Shown are the MAE and the standard deviation $\sigma$ of the AE per year for the 24-hour forecast horizon (96 time steps).

surpassed only by the observations. Surprisingly, in 2017, the simulated runoff had the highest relative importance of all years, although it featured the worst performance of the PBHM.

### 4.3.2 Sensitivity at peak runoff

To investigate the importance of the individual features at flood event runoff, we exclusively evaluated the integrated gradients for the two largest flood events of each year. Figure 9 shows the importances of the hindcast (left) and forecast (right) features for various distances of the predicted runoff peak to the forecast origin, $t_0$. In this regard, one means that the predicted peak is located at $t_{0+1}$ and analogously at $t_{0+96}$ for a value of 96.

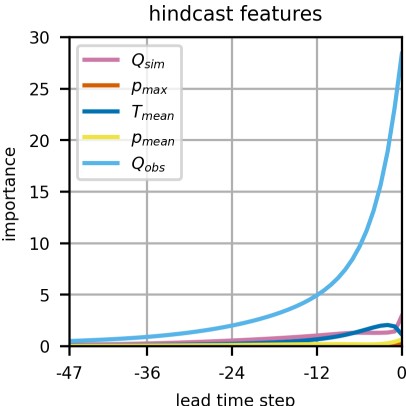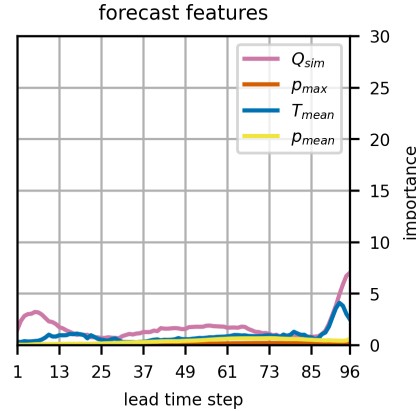

**Figure 8.** Importance of all input features summed over all testing years. Shown are the feature importances of the hindcast features (left) and the forecast features (right)

**Table 3.** Normalized feature importances per testing year. The values were normalized by the total sum of importance values per year. The most important input feature per year is highlighted in **bold**. Values less or equal to 0.01 are omitted to increase readability.

| year | hindcast features | | | | | forecast features | | | |
|---|---|---|---|---|---|---|---|---|---|
| | $Q_{sim}$ | $p_{max}$ | $T_{mean}$ | $p_{mean}$ | $Q_{obs}$ | $Q_{sim}$ | $p_{max}$ | $T_{mean}$ | $p_{mean}$ |
| 2013 | 0.09 | | 0.03 | | **0.55** | 0.19 | | 0.08 | 0.04 |
| 2014 | 0.08 | | 0.04 | 0.02 | **0.55** | 0.18 | | 0.08 | 0.04 |
| 2015 | 0.08 | | 0.08 | | **0.51** | 0.16 | | 0.12 | 0.03 |
| 2016 | 0.07 | | 0.07 | 0.02 | **0.49** | 0.18 | | 0.10 | 0.04 |
| 2017 | 0.06 | | 0.12 | | **0.32** | 0.24 | 0.02 | 0.16 | 0.07 |
| all folds | 0.08 | | 0.06 | | **0.51** | 0.19 | | 0.10 | 0.04 |

The results show that the closer the peak was located to the forecast origin, the more the forecast was influenced by the observed runoff. This comes as no surprise as the observed runoff at $t_0$ should be a reasonable predictor for the runoff at $t_{0+1}$. Interestingly, also in the case where the peak was located at the end of the forecast horizon $t_{0+96}$, the observed runoff still had a rather high impact on the forecast. As for the precipitation features, $p_{mean}$ and $p_{max}$, the importance of the former was found to be considerably higher. This means that the model gained more information from the precipitation volume than from

its intensity. Further investigating the mean precipitation feature revealed additional insights. First, the hindcast $p_{mean}$ showed to have a high influence when the peak was close to $t_0$, which then decayed exponentially with increasing distance of the peak runoff to the forecast origin. From a theoretical point of view, this makes sense, as some of the precipitation at this point has already passed the gauging station as surface runoff. Second, the forecast $p_{mean}$ showed little importance for forecasts that

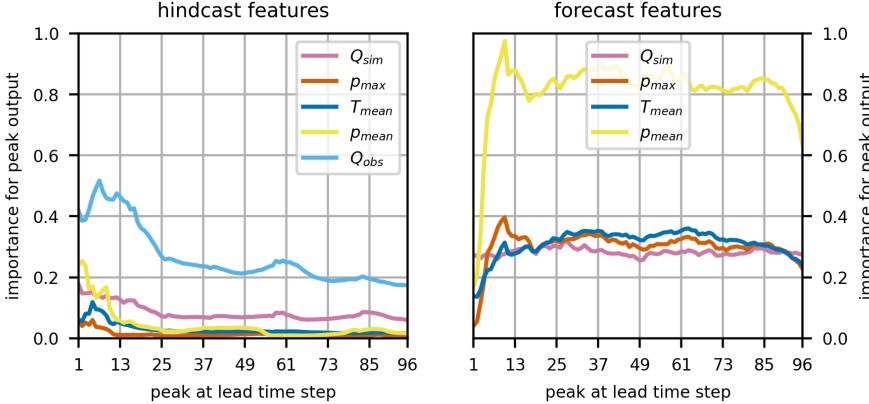

**Figure 9.** Importance of input features for the peak prediction in the forecast window summed over the two largest flood events per year. Shown are the feature importances of the hindcast features (left) and the forecast features (right).

were close to the forecast origin, but its importance showed to grow rapidly with increasing distance to $t_0$. This occurs as the rainfall needs time to concentrate and does not directly result in runoff. Also for predictions at flood event runoff, it showed that the PBHM-HLSTM relied on the PBHM's output. In the hindcast, it was found to be the second most important feature, while in the forecast its importance was found to be widely equal to that of the maximum precipitation $p_{max}$ and the mean temperature $T_{mean}$.

Table 4 summarizes the normalized feature importances for the two highest flood events per year, averaged over the hindcast and forecast periods, respectively. The results clearly show that the mean precipitation in the forecast period had the highest relative importance of all input features. In fact, it showed that the model mostly relied on forecast features for predicting flood event runoff, with the only exception being the observed runoff in the hindcast. As for the $Q_{sim}$, $p_{max}$, and $T_{mean}$ features in the forecast, all showed to have a more or less equal influence on the final flood event forecasts.

## 5  Discussion

In this study, we built upon the promising outcomes of prior research (see Rozos et al., 2021; Konapala et al., 2020; Frame et al., 2021) by exploring the potential of LSTMs for enhancing the forecast accuracy of PBHMs employed in operational flood forecasting systems. Following the approaches of Gauch et al. (2021) and Nevo et al. (2022), we developed our LSTM models using a hindcast-forecast architecture. This architecture was chosen as it facilitates an effective integration into operational forecasting systems. Specifically, the hindcast-forecast architecture allows for a clear separation between hindcast and forecast data, which comes with certain advantages. For example, this strategy would allow the model to distinguish between meteorologic forecasts and analyses, potentially enabling it to learn from their differences. Furthermore, the here presented cross-validation strategy enables a seamless continuous improvement of the model as new data becomes available. To assess

**Table 4.** Importance of features for the peak prediction in the forecast window. The values were normalized by the total sum of importance values per event. The most important input feature per event is highlighted in **bold**. Values less or equal to 0.01 are omitted to increase readability.

| year | event | hindcast features | | | | | forecast features | | | |
|---|---|---|---|---|---|---|---|---|---|---|
| | | $Q_{sim}$ | $p_{max}$ | $T_{mean}$ | $p_{mean}$ | $Q_{obs}$ | $Q_{sim}$ | $p_{max}$ | $T_{mean}$ | $p_{mean}$ |
| 2013 | 1st | 0.02 | | 0.02 | | 0.09 | 0.04 | 0.11 | 0.11 | **0.59** |
| | 2nd | 0.12 | | 0.02 | 0.04 | 0.20 | **0.28** | 0.02 | 0.12 | 0.19 |
| 2014 | 1st | 0.06 | | 0.02 | 0.02 | **0.32** | 0.22 | 0.02 | 0.11 | 0.23 |
| | 2nd | 0.07 | | 0.02 | 0.06 | 0.20 | **0.24** | 0.06 | 0.15 | 0.21 |
| 2015 | 1st | | | 0.03 | 0.03 | 0.05 | 0.05 | 0.12 | 0.18 | **0.52** |
| | 2nd | 0.02 | | | | 0.11 | 0.16 | 0.11 | 0.22 | **0.37** |
| 2016 | 1st | 0.02 | | | | 0.14 | 0.10 | 0.18 | 0.15 | **0.36** |
| | 2nd | 0.12 | | | 0.03 | 0.21 | 0.24 | 0.02 | 0.10 | **0.26** |
| 2017 | 1st | | | | | 0.02 | 0.07 | 0.27 | 0.16 | **0.44** |
| | 2nd | | | | | | 0.03 | 0.24 | 0.15 | **0.54** |
| all folds | 1st | 0.02 | | | | 0.12 | 0.10 | 0.17 | 0.14 | **0.41** |
| | 2nd | 0.06 | | | 0.03 | 0.13 | 0.17 | 0.10 | 0.15 | **0.34** |
| | both | 0.04 | | | 0.02 | 0.12 | 0.13 | 0.14 | 0.14 | **0.38** |

the benefits of the LSTM-based forecasts, we developed two LSTM model variants and compared their forecast skills to that of a conventional ARIMA model, using one underperforming PBHM as a case study. To ensure comparability between the LSTM and ARIMA approaches, one LSTM (eLSTM) was restricted to use the same data as ARIMA, while the other incorporated additional meteorologic variables (PBHM-HLSTM). Of particular interest was how the LSTM approach improved prediction accuracy, especially at flood event runoff and for longer lead times — both being recognized weaknesses of ARIMA for cases where the underlying PBHM provides poor initial estimates.

When comparing the forecasts obtained by the LSTM and ARIMA models, we observed that both methods had certain advantages and disadvantages. ARIMA generally demonstrated a very high accuracy in the first forecast steps. However, this initial accuracy often showed to decline quickly with increasing lead time. These findings align with those presented in previous studies such as Brath et al. (2002) or Broersen and Weerts (2005). In contrast, the LSTMs generally exhibited a larger error in the first steps but were able to mostly sustain their initial accuracy over the 24-hour forecast horizon. This became particularly evident when observing the variance of the absolute errors. Both LSTM models, particularly the PBHM-HLSTM, displayed a considerably lower error variance compared to the results obtained by ARIMA. This suggests that they produced exceptionally poor forecasts less often. Interestingly, ARIMA performed exceptionally well in terms of PBIAS. The reason for that was

found in ARIMA's high accuracy for forecasts that followed a clear trend or pattern, which in hydrologic model applications occurs most often during baseflow conditions.

When focusing solely on the forecast skills at flood event runoff, the LSTMs clearly outperformed ARIMA. This became
particularly evident when investigating the models' timing errors, i.e., the temporal offset between the maximum observed and simulated peak runoff. While both LSTM variants were able to significantly reduce the initial timing errors of the PBHM, this was not achieved by ARIMA. This implies that ARIMA was not able to adequately transform the event hydrographs in instances where the underlying PBHM was not able to give an adequate initial estimation, a fact that was also shown by Liu et al. (2015). As for the magnitude errors, i.e., the difference between the maximum observed and simulated runoff, only the
PBHM-HLSTM was able to achieve somewhat satisfying results. Interestingly, the eLSTM even performed worse than ARIMA in this regard. This indicates that the eLSTM did not receive sufficient context from the observed and simulated runoff alone to accurately capture the magnitude of flood events. This underscores the importance of incorporating meteorologic variables when employing LSTM models in operational forecasting systems.

Considering the comparably high performance of the PBHM-HLSTM in this study and more generally the remarkable
capabilities of LSTM models in predicting river runoff (e.g., Kratzert et al., 2019b), it raises the question regarding the added benefits that the underperforming PBHM provides. To assess the added value of the PBHM in this study, we evaluated the relative importance of each of the PBHM-HLSTM's input features. Our findings indicate that, on average, the PBHM-HLSTM model heavily relied on the results of the PBHM, particularly its forecasts. In fact, the PBHM's runoff predictions were identified as the second most important feature, following the observed runoff. While the PBHM-HLSTM at flood event runoff
did to some extent also rely on the PBHM's forecasts, the mean catchment precipitation emerged as the most important feature in these instances. These findings also explain the large performance gap between the eLSTM and the PBHM-HLSTM at flood event runoff.

When employing forecast-enhancing models in operational flood forecasting systems, several important considerations must be taken into account. First and foremost, such models are no all-in-one device suitable for every purpose. Although the
435 here presented PBHM-HLSTM was able to significantly improve upon the PBHM's forecasts it is still a post-processing technique that is meant to enhance predictions at the specific location of the gauging station, while leaving the PBHM's system states untouched. However, often these system states, e.g., the state of the snow cover, the soil moisture, or spatially distributed information of the runoff, function as an additional decision criterion for the system's operator and are often used for implementing more complex forecasting chains. Considering the poor performance of the PBHM in this study, its system
states are most likely not correct and can thus not provide any added benefit. Furthermore, it has to be considered that there is a reason why the PBHM's performance is poor. Often this can be linked to poor model parametrization, the inability of the model to capture some important catchment processes, or uncertainties in the input data. For the latter, these uncertainties might be present in the data used for setting up the PBHM, in the meteorologic forcings, or in the data used for calibration (e.g., the gauge runoff). Notably, in contrast to PBHM's, data-driven models (e.g., LSTMs) might be adept at learning any systematic
errors embedded in the data, consequently improving forecast accuracy. Overall, we believe that data-driven forecast-enhancing

strategies are highly valuable in contexts like the one presented in this study, where the PBHM alone fails to deliver satisfactory forecasts.

Although the PBHM-HLSTM model presented here already achieved a comparably high forecast accuracy, there exists potential for future enhancements. First, refining the pre-processing phase, especially through more targeted feature engineering could further enhance the model's predictive capabilities. Second, the target data (gauge runoff) can be diagnosed. For instance, adopting the probe technique presented by Lees et al. (2022) could be used to identify behavioral anomalies in the LSTM cell states by comparing multiple catchments. Lastly, future work could also focus on investigating a hybrid ARIMA-LSTM approach, potentially further increasing the model's prediction accuracy, particularly in the first forecast steps.

## 6 Conclusions

In this study, we explored the potential of Long Short-Term Memory (LSTM) networks as a post-processing strategy for enhancing the forecast performance of an underperforming process-based hydrologic model (PBHM). We specifically compared this post-processing strategy to a conventional AutoRegressive Integrated Moving Average (ARIMA) model, as such models are often employed in operational flood forecasting systems. Our focus was on the models' performances for extended lead times and particularly at flood event runoff, both being critical aspects in operational flood forecasting. To facilitate an objective comparison, we developed two LSTM model variants. One variant, eLSTM, was restricted to use the same input data as ARIMA, namely observed runoff and the runoff generated by the PBHM, while the other, PBHM-HLSTM, additionally incorporated meteorologic variables. Furthermore, we assessed the added value of the underperforming PBHM's results on the predictions of the PBHM-HLSTM by evaluating the importance of each of the model's input features. The main findings of this study can be summarized as follows:

– All model variants (ARIMA, eLSTM, and PBHM-HLSTM) significantly enhanced the forecast accuracy of the existing PBHM.

– ARIMA achieved a particularly high accuracy in the first forecast steps. However, this initial accuracy declined quickly with increasing lead time. In contrast, the LSTM models showed a larger initial error but mostly maintained their initial accuracy over the 24-hour forecast horizon.

– ARIMA showed shortcomings in forecasting flood event runoff. Specifically, it failed to accurately predict the timing and the maximum peak runoff of the flood events. The eLSTM improved timing predictions but significantly underestimated the magnitude of the events. Only the PBHM-HLSTM was able to sufficiently predict both the timing and the magnitude of the flood events.

– Despite the PBHM's poor performance, the PBHM-HLSTM still considered its output informative. On an annual average, the PBHM's output was found to be the second most important feature, following the observed runoff. For flood event predictions the PBHM's results were also found to be important, but the catchment's mean precipitation was identified as the most critical input feature in these cases.

**Table A1.** Statistics of the catchment's runoff (gauge observation $Q_{obs}$, PBHM simulation $Q_{sim}$) as well as its mean precipitation, maximum precipitation, and temperature ($p_{mean}$, $p_{max}$ and $T_{mean}$).

| | | | year | | | | | | |
|---|---|---|---|---|---|---|---|---|---|
| parameter | statistic | unit | 2011 | 2012 | 2013 | 2014 | 2015 | 2016 | 2017 |
| $Q_{obs}$ | $\mu$ | $\mathrm{m^3 s^{-1}}$ | 0.57 | 1.01 | 1.21 | 1.17 | 0.71 | 0.83 | 0.57 |
| | $\sigma$ | $\mathrm{m^3 s^{-1}}$ | 0.25 | 0.84 | 0.68 | 0.67 | 0.33 | 0.68 | 0.23 |
| | max | $\mathrm{m^3 s^{-1}}$ | 9.61 | 25.20 | 15.00 | 7.27 | 5.85 | 17.90 | 9.21 |
| | $\Sigma$ | $\mathrm{hm^3}$ | 18.0 | 31.9 | 38.1 | 36.8 | 22.4 | 26.2 | 17.7 |
| $Q_{sim}$ | $\mu$ | $\mathrm{m^3 s^{-1}}$ | 0.62 | 1.10 | 1.40 | 1.35 | 0.76 | 0.92 | 0.91 |
| | $\sigma$ | $\mathrm{m^3 s^{-1}}$ | 0.33 | 0.83 | 1.02 | 0.72 | 0.52 | 0.70 | 0.48 |
| | max | $\mathrm{m^3 s^{-1}}$ | 4.20 | 8.94 | 11.40 | 7.68 | 4.50 | 6.43 | 4.61 |
| | $\Sigma$ | $\mathrm{hm^3}$ | 19.5 | 34.9 | 44.1 | 42.6 | 23.8 | 29.0 | 28.3 |
| $p_{max}$ | max | $\mathrm{mm\,h^{-1}}$ | 118 | 180 | 100 | 84.6 | 109 | 231 | 173 |
| $p_{mean}$ | max | $\mathrm{mm\,h^{-1}}$ | 29.2 | 69.7 | 45.8 | 33.5 | 38.6 | 61.6 | 69.3 |
| | $\Sigma$ | mm | 871 | 1289 | 1284 | 1225 | 912 | 1188 | 1153 |
| $T_{mean}$ | $\mu$ | °C | 6.88 | 6.72 | 6.43 | 7.47 | 7.56 | 6.98 | 6.94 |
| | $\sigma$ | °C | 7.97 | 8.72 | 8.21 | 6.72 | 7.90 | 7.59 | 8.27 |

To summarize, in this study we demonstrated that LSTM models can pose a viable alternative to frequently employed ARIMA correction models in operational flood forecasting systems.

# Appendix A: Statistics of the input data

## A1   Model input data

Table A1 shows the key statistics of the observed and simulated runoff as well as the meteorologic forcings used in this study.

## A2   PBHM model residuals

Two statistical tests have been employed to analyze the PBHM's residuals. First, the goodness of fit test was used to analyze how close the residuals follow a Gaussian distribution. For this purpose the Filliben $r$ correlation value (Filliben, 1975) was computed, for which a value close to one signifies a Gaussian distribution. Second, the Lagrange multiplier statistic of the Breusch-Pagan Test (Breusch and Pagan, 1979) was evaluated to assess the degree of heteroscedasticity of the residuals. The critical value for homoscedasticity was computed for a 5 % significance as 3.84 based on the Chi-distribution given one degree of freedom. The application of the Box-Cox transform, using a $\lambda$-value of 0.2, showed an increase in Gaussianity in the

residuals' distribution as well as a reduction of heteroscedasticity, even below the critical value for homoscedasticity. Fig. A1 shows a QQ-plot including the Filliben $r$ test statistics for the original and Box-Cox transformed residuals (left), alongside a scatterplot of the PBHM's residuals against the observed runoff, which includes the test statistics of the Breusch-Pagan test (right).

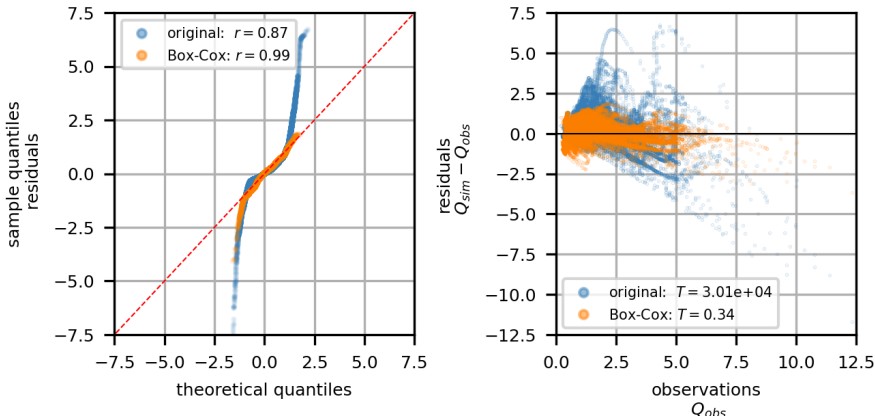

**Figure A1.** QQ-plot with Filliben $r$ test statistics for the original and Box-Cox transformed residuals (left) Scatterplot of the PBHM's original and transformed residuals against the observed runoff including the test statistics of the Breusch-Pagan test (right).

## A3    Autocorrelation evaluation of the PBHM residuals

We evaluated the Autocorrelation Function (ACF) and Partial Autocorrelation Function (PACF) for the PBHM's residuals. Both are visualized in Fig. A2. The correlation values and their 5 % significance bounds were obtained by bootstrapping, where the residuals were analyzed for each year and the results averaged. Fig. A2 includes the original PBHM residuals, the Box-Cox transformed residuals as well as the residuals following one differentiation operation.

## Appendix B: Evaluation metrics

**B1    Nash-Sutcliffe Efficiency (NSE)**

The Nash-Sutcliffe Efficiency (NSE, Nash and Sutcliffe, 1970) quantifies how well the model performs compared to a simple mean runoff benchmark. In its original form, the NSE can be written as:

$$NSE = 1 - \frac{\sum_{t=1}^{N}(Q_{obs,t} - Q_{sim,t})^2}{\sum_{t=1}^{N}(Q_{obs,t} - \overline{Q}_{obs})^2} \tag{B1}$$

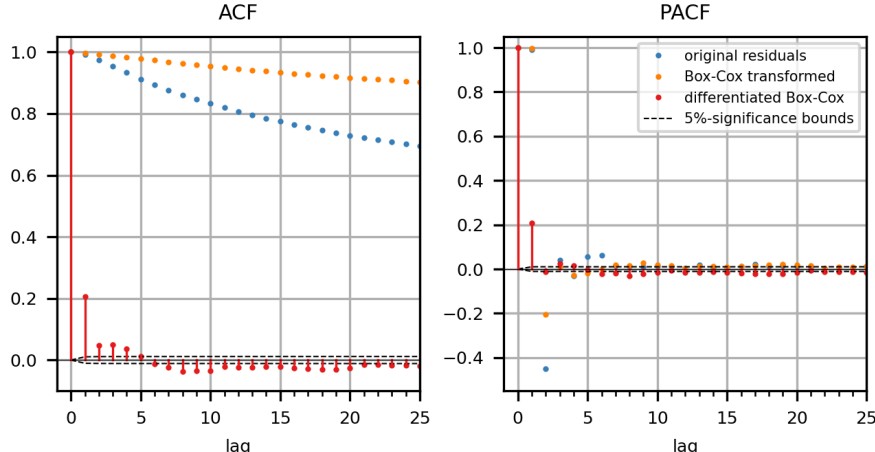

**Figure A2.** Autocorrelation Function (ACF, left) and Partial Autocorrelation Function (PACF, right) for the original PBHM model residuals, the Box-Cox transformed residuals as well as the residuals following one differentiation operation.

where $Q_{obs,t}$ and $Q_{sim,t}$ is the observed and predicted runoff, respectively. The NSE is bound between 1 and $-\infty$, with 1 indicating perfect model predictions.

## B2 Kling-Gupta Efficiency (KGE)

The Kling-Gupta Efficiency (KGE) was proposed by Gupta et al. (2009). It is a combined efficiency metric that considers the correlation, the bias, and the variability of the flow. In this study, we utilized the modified Kling-Gupta Efficiency (Kling et al., 2012), which can be written as:

$$KGE = 1 - \sqrt{(r-1)^2 + (\beta - 1)^2 + (\gamma - 1)^2} \tag{B2}$$

where $r$ is the correlation term, $\beta$ is the bias term given by the ratio of the mean of the simulated and observed runoff values $\mu_{sim,t}/\mu_{obs,t}$ and $\gamma$ is the variability term, which is computed from the standard deviations and the mean values as $\frac{\sigma_{sim,t}/\mu_{sim,t}}{\sigma_{obs,t}/\mu_{obs,t}}$. The KGE is bound between 1 and $-\infty$, with 1 indicating perfect model predictions.

## B3 Percent bias (PBIAS)

The PBIAS is a measure that quantifies if the model tends to underpredict or overpredict the observed runoff. It can be written as follows (Yilmaz et al., 2008):

$$PBIAS = \frac{\sum_{t=1}^{N}(Q_{sim,t} - Q_{obs,t})}{\sum_{t=1}^{N} Q_{obs,t}} \cdot 100 \tag{B3}$$

where $Q_{sim,t}$ and $Q_{obs,t}$ is the observed and predicted runoff, respectively. The PBIAS can take both positive and negative values, where positive values indicate that the model on average overpredicts the observations and vice versa. A PBIAS close

to zero indicates a widely unbiased model.

## B4 High-segment volume percent bias (FHV)

The FHV quantifies high flows with an exceedance probability lower than 0.02 based on the flow duration curve (Yilmaz et al., 2008). It can be written as follows:

$$FHV = \frac{\sum_{i=1}^{H}(Q_{sim,i} - Q_{obs,i})}{\sum_{i=1}^{H}(Q_{obs,i})} \cdot 100 \tag{B4}$$

where $Q_{obs,i}$ and $Q_{sim,i}$ is the observed and predicted runoff, respectively, and $i = 1, 2, \ldots H$ is the index of the flow value located within the high-flow segment of the flow duration curve.

## Appendix C:  Auxiliary information on LSTM hyperparameter tuning

For tuning the hyperparameters, we selected a combined objective function $f_{obj}$ consisting of the NSE and KGE metrics. The objective function was computed as follows:

$$f_{obj} = 2 - KGE - NSE \tag{C1}$$

where zero would indicate a perfect fit by the model.

Table C1 shows the LSTM hyperparameters subjected to optimization, their search space, and their final values after tuning. Additionally, we investigated two different hindcast lengths, namely 12 and 24 hours, and chose the final model variants based on the lowest objective function value.

Figure C1 depicts the train and validation losses per epoch for all five fold models and selected model variants. The tuner used an early stopping mechanism by monitoring the development of the validation loss.

## Appendix D:  Model predictions for the largest flood event per year

Figure D1 shows the location and magnitude of the estimated flood peak for all 96 lead time predictions for the largest flood event per year. Cumulative average precipitation over the catchment and the PBHM's predictions are given as a reference. It

can be seen, that the predicted peaks of the PBHM-LSTM model incorporating information on precipitation and temperature during the forecast horizon, matched more closely to the actual peaks than the predictions from the variants solely built on the PBHM's results and the observed runoff. A summary of these findings can also be found in Table 2.

**Table C1.** Hyperparameter tuning information, defined search space, and final parameter set for the LSTM models.

| Parameter | Search space | | eLSTM* | eLSTM-H48 | BPHM-HLSTM-H96 | BPHM-HLSTM* |
|---|---|---|---|---|---|---|
| | Min. | Max. | | | | |
| ID best trial | | | 41 | 40 | 48 | 22 |
| Objective | | | **0.232** | 0.239 | 0.169 | **0.162** |
| N. of LSTM units | 4 | 32 | 17 | 22 | 20 | 23 |
| Initial learning rate | 1e-3 | 1e-2 | 0.00774 | 0.0090 | 0.0065 | 0.00912 |
| Dropout rate | 0.01 | 0.5 | 0.247 | 0.0228 | 0.0633 | 0.039 |
| Batch size | | | 4000 | 4000 | 4000 | 4000 |
| Retrain epochs | | | 5 | 5 | 5 | 5 |
| Hindcast length | | | 96 | 48 | 96 | 48 |

*selected for further processing based on tuner objective

*Code and data availability.*  The Python code and processed data presented in this study are stored on **Zenodo** (Gegenleithner et al., 2024b). The published data was derived from the following datasets (I) Gauge runoff: Styrian Government, Department 14 – Water Management, Resources and Sustainability (Hydrographic Service of Styria). The data was validated by the provider. The time stamps were converted from GMT+1 to UTC by the authors. (II) Meteorologic data: The meteorologic data was provided by GeoSphere Austria. More specifically, 1x1 km rasters were provided from which we extracted catchment averaged values. Those averaged values are included in the dataset. (III) Hydrologic modeling results: The hydrologic modeling results were obtained from Gegenleithner et al. (2024a). The developed Python code is also available on **GitHub**.

*Author contributions.*  Sebastian Gegenleithner: Conceptualization, Methodology, Data curation, Writing - original draft preparation. Manuel Pirker: Conceptualization, Methodology, Data curation, Writing - original draft preparation. Clemens Dorfmann: Funding acquisition, Writing - review & editing. Roman Kern: Writing - review & editing. Josef Schneider: Supervision, Writing - review & editing

*Competing interests.*  The authors declare that they have no conflict of interest.

*Acknowledgements.*  We express our gratitude to the Styrian Government, Department 14 – Water Management, Resources and Sustainability (Hydrographic Service of Styria) and to GeoSphere Austria for providing the data for this study.

We declare that during the preparation of this work we used generative AI to enhance specific sections of the written content. The content was reviewed and we take full responsibility for the quality of this publication.

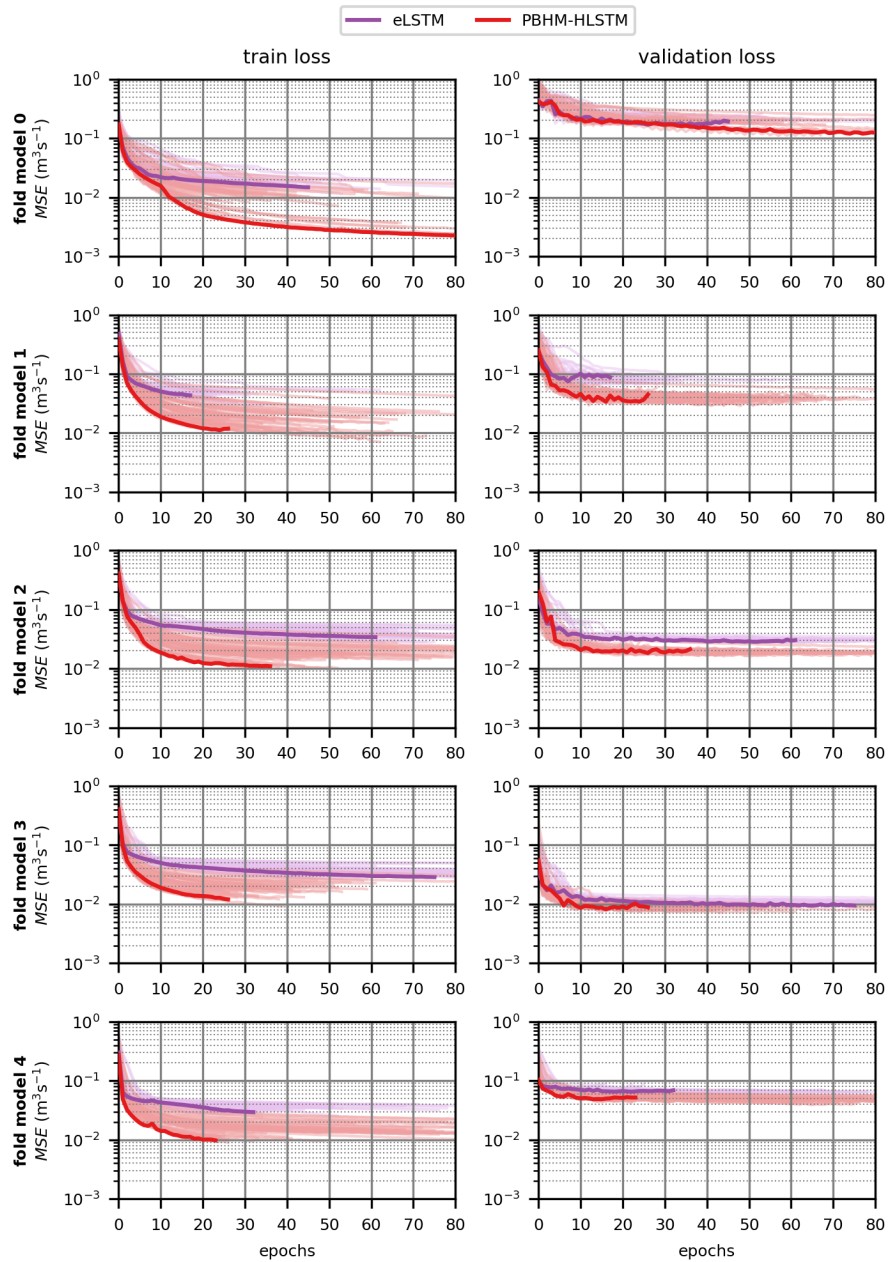

**Figure C1.** Best models' losses during training and validation.

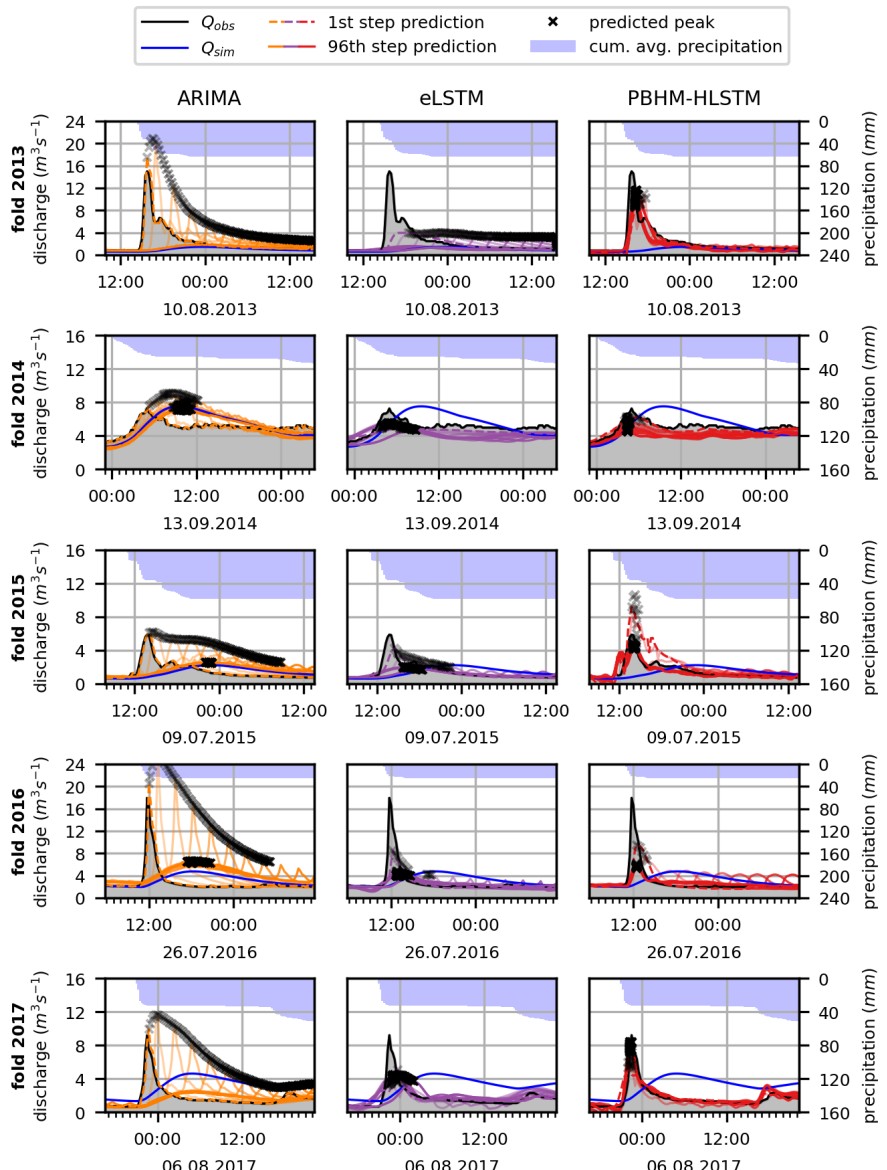

**Figure D1.** Forecast comparison for the largest runoff event per year. Given are the results of the ARIMA, eLSTM, and PBHM-HLSTM models.

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
