# Peer review of "Long Short-Term Memory Networks for Enhancing Real-time Flood Forecasts: A Case Study for an Underperforming Hydrologic Model"

_EGUsphere, 2024_

## Author Response (AR1)

**Author's comments to referees**

We express our gratitude to the referees as well as the handling editor for their valuable time in helping to improve this manuscript. In our opinion the referees' valuable input considerably improved the revised manuscript. This document will lay out all major changes we added to the initial manuscript. To improve readability, we included the referees' comments (gray boxes) and describe our changes for each of the comments. Our initial response and planned activities are also included in the document, which were all implemented. The major changes we added to the manuscript can be summarized as follows:

- We added an intermediate model between ARIMA and the PBHM-HLSTM, named eLSTM. This model was trained with the same data as ARIMA and implemented with the same architecture as the PBHM-HLSTM
- We completely removed the HLSTM model, which was implemented to check if the simulated runoff was useful for the LSTM model. Instead we conducted a sensitivity analysis based on the integrated gradient method.
- A more in-depth statistical analysis of the PBHM model residuals was added, which was also used to majorly rework the ARIMA model development section
- We made the fact that ARIMA and the LSTM models in principle have different underlying principles for obtaining their forecasts more clear throughout the manuscript. ARIMA forecasts residuals, while the LSTM models directly predict runoff. This was addressed accordingly in the relevant sections of the manuscript. Specifically, we changed the title and also made this fact clear throughout all other parts of the manuscript.
- We majorly reworked large parts of the manuscript, i.e., the title, the research questions, etc., to properly address the comments of the referees.
- We also added all minor changes suggested by the referees as states below.

**AC Comment to RC1 – Implemented changes**

Dear editor and authors,

The following comment details my review of the manuscript "Long Short-Term Memory Networks for Real-time Flood Forecast Correction: A Case Study for an Underperforming Hydrologic Model" submitted to HESS.

In this preprint the authors present a model comparison study in which two (or three depending on the application) models are compared in their ability to forecast runoff. The models compared are all statistical- or machine learning-based models which take as inputs predictions of an underperforming conceptual model. The preprint is well written and the results are compelling. The scope of the manuscript is well suited for HESS and it has potential to be a great contribution to the literature on runoff forecasting, as well as models which combine physics-based and data-driven approaches.

However, I have a number of major and minor comments/suggestions that should be addressed before final publication and ultimately will benefit the manuscript and overall study.

**Major Comments**

*The comparison is not "fair"*

What the ARIMA model is doing is very different than what the LSTM-based models are doing and this "unfair" comparison is apparent in the results. Evidently the model which is able to use

data from precipitation in its forecasting step will be better at predicting events that have precipitation as its main driver and not the current or past discharge as calculated by an underperforming PBHM.

What is missing is a model that is in-between the ARIMA and HLSTM-PBHM and bridges the gap between the two approaches. In principle this could be an ARIMA which considers exogenous inputs (ARIMAX) or an LSTM which predicts errors without the aid of external variables for a direct comparison with the presented ARIMA model. This way we see how performance changes from having a model that is only correcting the PBHM (ARIMA), to a model which relies in the PBHM but can use other inputs when the PBHM fails (ARIMAX), to a model that accounts for all available input data and chooses what to use (HLSTM-PBHM).

Ultimately, I think the models which should be part of the study are: ARIMA, ARIMAX, LSTM which predicts errors only using $Q_{sim}$ and $Q_{obs}$ (name: $_eLSTM$?), and the presented HLSTM-PBHM.

To ensure comparability, we included an additional model variant, eLSTM, as suggested by the referee. As for the ARIMA model with exogenous variables suggested by the referee, we did not achieve a higher model performance compared to the original ARIMA model. Therefore, this model variant was not included. However, we do believe that the eLSTM serves well as an intermediate model between ARIMA and the PBHM-HLSTM, as it utilizes the same data as ARIMA and the same architecture as the PBHM-HLSTM. We do believe that we could utilize this additional model well throughout the manuscript to highlight important differences between the ARIMA and LSTM model strategies.

Furthermore, although the HLSTM was added to address a specific concern regarding the combination of the PBHM and an LSTM, I don't think its able to address this issue effectively as the authors also recognize by saying that in their findings: "We did not find strong evidence of whether the inclusion of the PBHM's results benefited the accuracy of the LSTM." My suggestion is that the HLSTM is completely dropped. Give that this model simply serves to check if $Q_{sim}$ is somewhat informative to the LSTM in the HLSTM-PBHM, this could be made clearer through a sensitivity analysis and not using a different model with also a different architecture. I suggest the sensitivity analysis could be done using integrated gradients as Kratzert et al. (2019) or simply by replacing the input of $Q_{sim}$ for noise and seeing the effect it has on the predictions by the model. If the LSTM does not consider the input from $Q_{sim}$ useful, there should be no effect by replacing this input for noise and vice versa.

This was an excellent point of the referee and we have put much thought into this topic. We ultimately decided to completely remove the HLSTM model (it was more or less replaced by the eLSTM) and performed a sensitivity analysis instead. As for the method, we decided to use integrated gradients, as suggested by the referee. In accordance to the overall structure of the manuscript, we divided the IG evaluation into an overall evaluation and one only conducted for flood event runoff, which in our opinion gave some interesting insights to the model. In short, it showed that the PBHM's simulations had some impact on the model, especially when averaged over the entire year.

This flaw in the design of the study is also reflected on the research questions established in the introduction. None of the research questions concern the ARIMA model, so why is it part of the study at all? From the point of view of the RQs, the study should only focus on the HLSTM-PBHM and the previously described *e*LSTM which could be considered a deep-learning adaptation of ARIMA while the HLSTM-PBHM is more akin to an ARIMAX, keeping the scope of

> the paper within error correcting strategies, and then the discussion can focus on the benefit of precipitation as an input during forecasting, and the difference between years where the PBHM is acceptable (2014 and 2016) in contrast to when it's terrible (2017).

When we designed the layout of this study, we intended to compare a classical ARIMA model to LSTM models for forecast corrections. We do believe that this will be interesting for readers of HESS, as currently still many operational forecasting systems utilize ARIMA correction strategies. However, we wanted to show an alternative for instances where ARIMA models do not work well due to their linear nature. Therefore, we also selected a particularly poor performing PBHM to conduct this study. We have now restated the research questions of this study to more closely reflect our initial intentions, namely "how much better are LSTM models in improving the forecast accuracy in instances where ARIMA might not work". We also improved many other sections in this regard.

> *Hyperparameters of the LSTM-based models*
>
> In the supplemental information of the paper by Nevo et al. (2022), their model is described to have an LSTM of 128 hidden units for hindcast and another 128 hidden units for forecast, which is similar to the 96 used in this article, but in the case of Nevo et al. (2022), the model was trained to forecast in at least 165 basins which use LSTM for the "stage forecast model". The architecture presented by Nevo et al. (2022) is also based on the MTS-LSTM presented by Gauch et al. (2021). Although the purpose of that second paper is not forecasting, the idea of "handing" the hidden states from one LSTM to another is the same, and in their case both LSTMs which send and receive the hidden states have 64 hidden nodes. This is also applied in a regional case study in which the amount of data that the model needs to ingest is a lot larger. Finally, in a more recent example by some of the same authors of the previous papers, Kratzert et al. (2024) train single-basin LSTMs using models with hidden nodes ranging from 8 to, at most, 32. This is not a criticism of not using LSTM in a regional setting, in my view LSTMs are still valid for application in a single basin, acknowledging their limitations, but their size shouldn't be the same as those used in regional modelling.
>
> This could be addressed by adding smaller sizes into the hyperparameter search space and I would encourage the authors to present their results for training/validation in supplemental material of the article in the form of loss curves, metrics in training/validation, etc.

We agree with referee 1 that the number of nodes should scale with the complexity of the task at hand, i.e., a regional model should be less complex than for cases with multiple basins. We retrained our models with fewer hidden nodes. Additionally, we added the training and validation losses in the appendix. We also investigated one additional hindcast length of 48 (12 hours) for both model variants, as the evaluation of the IG on the original model from the preprint showed, that there was little to no information drawn from inputs more than 12 hours in the past. For the PBHM-HLSTM this led to a better model (based solely on the tuner objective score), for the eLSTM the hindcast length of 96 scored better. For the paper we then used the better model of each variant.

> Checking the code repository and additional files provided by the authors, the '\tb' folder was not included so the TensorBoard logs cannot be checked.
>
> On checking the code further, there are remainder classes and functions that were not part of the study such as the models that include 'CNN' layers. I would suggest a general cleanup, but

> the code definitely runs and I was able to generate one of the trials done for the study and the corresponding logs.

We included the TensorBoard logs as well as a cleaned up-version of the code.

> Taking a look at some of the post-processing notebooks, I find the information presented in the `notebook_aux_peak_events.ipynb` to be informative and would encourage the authors to include some of the plots for peak events in section 4.2 in the manuscript. The colors for the ARIMA model need to be adjusted though, because they are the same as the "measured" data.

As the referee suggested, we included a plot of the peak runoff predictions. These plots were included in the appendix. We also changed the colors and the general layout of the plot.

**Minor Comments**

> General: I suggest that the name of the HLSTM-PBHM should be flipped around to PBHM-HLSTM as the PBHM is the initial step in the pipeline.

We changed the name of the LSTM model to PBHM-HLSTM in the manuscript.

> Line 72: The hindcast-forecast LSTM cannot be called "novel" as it is adapted from the approach of Nevo, et al. (2022), which in-itself is adapted from other sources.

We agree. We rephrased the relevant parts in the manuscript and also added the paper Gauch et al. (2021) as a reference.

> Line 146: How was the search-space for the hyperparameters of the ARIMA model defined?

The search space for ARIMA was defined based on prior testing. Now we determined the order of the ARIMA model by using the PACF and ACF functions as a starting point. And iteratively determined the order by monitoring the results, whilst checking for overfitting. Additionally, we monitored the ARIMA model residuals. This is also in alignment with the comment by referee 2.

> Line 171: Why was a loss function which combines two metrics chosen? Was minimizing NSE or simply MSE tried? From Appendix B the authors say that this was adapted from Nevo et al. (2022) but in that paper they minimize a negative log-likelihood as they have a probabilistic model.

We fixed the citation of Nevo et al. (2022) and again apologize for the inconveniences. We also switched the loss function to a simple MSE.

> Figure 4: On the right-hand side of the figure I'm missing the $Q_{forecast}$ as in Figure 3. Further the "Targets" should be dropped from this figure or added to Figure 3 to have somewhat equivalent descriptions of the model architectures.

This issue was fixed.

> Table 2: Although PBIAS is a good overall metric to include, these results would greatly benefit of including metrics which directly target low- and high-flows like FHV and FLV. See Gauch et al. (2021) or directly Yilmaz et al. (2008). Some of the results described in the text using PBIAS are more suited to be described using these two metrics.

This is a great point. We included the FHV bias and also tested the FLV bias. However, for the FLV we did not get meaningful results. The reason for that was that the catchment in this study

had a very low base runoff (around 1 m³/s) and in winter even lower. On a few occasions, the LSTM models thus produced runoff predictions close or equal to zero. This resulted in "exploding" FLV values. For this reason, we kept the PBIAS, which in our opinion is a good indicator of the overall bias and included the FLV bias as suggested by the referee.

> Also, I find the reported PBIAS metrics for the ARIMA models to be strange given their KGE and NSE per year. From the code, in the `run_arima.py` I see it calls the `calculate_bias` function but only prints values to the screen while in `notebook_tab1-4_table_results.ipynb` the metrics are read directly from a file. I'm guessing that `metrics.txt` is generated using the data in the other files in that folder `metric_nse.txt`, etc. but I also don't see where in the code those files are dumped. `run_arima.py` dumps the `all_fc_df` as a pickle, but I'm not sure if that DataFrame is used to generate the metrics `.txt` files.

Yes, we were also surprised that the BIAS error of the ARIMA model was that low. However, upon further investigation, it showed that ARIMA produced many "perfect" forecasts throughout the year. This was especially true in baseflow conditions, which had a large positive impact on the PBIAS values. Noteworthy, this is also the reason why ARIMA in some instances has quite good KGE – KGE has a direct term for the BIAS - statistics, whilst the NSE is not that good.

However, we restructured the code to allow for a more transparent evaluation of the individual models. Specifically, we created a separate notebook, namely 'pre_evaluate_metrics.ipynb', which evaluates the metrics based on the models' forecasts.

> Table 3: Generalization refers to a models ability to predict in previously unseen data drawn from the same distribution as the one used to train the model. In doesn't have to do with differences between validation and testing. I would consider correcting this table with differences between the testing and training sets, or not including it at all as most of the discussion cantered around these results appears or can be included in other sections.

We agree with the referee and have completely removed all parts that previously discussed "generalization capabilities".

> Fig. 6 and Fig. 7: The legend of both columns should not be shared. Currently it appears as if the "normalized win ratio" has something to do with the standard deviation of each model.

We agree with the referee that this part was confusing. We completely removed the "normalized win ratio", as it added no informative added value. Instead, we went for the simple standard deviation, which can be interpreted more easily and in our opinion is able to better convey the results.

**AC Comment to RC2 – Implemented changes**

> This study proposes an application of Long Short-Term Memory Networks (LSTM) complemented by the results of a hydrological model (PBHM) for operational flood forecasting in a smaller mountainous catchment. The performance of the resulting HLSTM-PBHM is compared with an ARIMA error correction model and a standalone application of the LSTM (HLSTM).
>
> The results of this study are particularly significant, as they reveal performance improvements for the HLSTM-PBHM, especially for larger lead times. These findings have practical implications for flood forecasting in similar catchments.

The paper is within the scope and very interesting for the readers of HESS. The authors address a topic of high relevance for flood forecasting since studies focusing on small catchments and requiring sub-daily time steps are limited.

The authors have done a commendable job of presenting the scientific results concisely and well-structured. However, I have some fundamental comments on the interpretation of the proposed method and the concept of the experimental design to compare the different approaches:

From my perspective, the proposed HLSTM-PBHM is an informed approach that uses precalculated results of the hydrological model (PBHM) combined with observations for the hindcast rather than applying an explicit error correction as the ARIMA error correction model does. Therefore, the title of the paper should reflect this, and I suggest revising it.

We agree with the referee's comment that, in principle, the ARIMA and LSTM models follow a different methodology of how the final forecasts are obtained. We revised the title and all sections of the manuscript accordingly.

This approach's consequence is that the input data used and the internal corrections of HLSTM-PBHM cannot be compared with the residual errors of the hydrological model and the corrections calculated by the explicit error correction models.

A general shortcoming of comparability between ARIMA and the HLSTM-PBHM was also pointed out by referee 1. We addressed this issue by introducing an intermediate model, eLSTM. As for the residual errors, such a comprehensive analysis as presented in the literature by the referee was not possible, as it would have exceeded the scope of the paper. We ultimately decided to address model residuals only and briefly for the ARIMA model, as it is an important quality criterion for this model type, as also shown in the literature presented by the reviewer. For the final model comparisons, we still relied on comparing the forecast quality, whilst making clear that ARIMA and the LSTM models operate on different principles.

In general, a comprehensive analysis of the residual errors of the PBMH model, e.g., the underlying statistical distribution, would be helpful and give the reader more insight to interpret the results. It would also prove the assumption of whether the errors are normally distributed. Many studies (among them [1]) found a high heteroscedasticity variance of residuals, which should be checked and considered for the residuals in the study.

A comprehensive analysis of the PBHM's residuals was added to the appendix, as we thought it might interrupt the readability of the main body. As the reviewer correctly pointed out, also our residuals did show a high degree of heteroscedasticity in the residuals, which could be stabilized by applying a Box-Cox transformation as suggested by the referee. The transformation also improved Gaussianity. The statistics of the PBHM's residuals, however, have resulted in a major revision of the ARIMA model section.

Please also briefly introduce the PBHM model in the Methods Section as it is used in the study.

We gave some more insight to the hydrologic model used in the study cited.

Multiple errors exist in flood forecasting due to meteorological uncertainties and those rising from the structure and parametrization of the hydrological model. Please elaborate on how the

different contributions could be considered in future developments of HLSTM-PBHM in the discussion.

We agree with the referee that a variety of uncertainties exist in operational flood forecasting (e.g., meteorology, streamflow observations and all uncertainties concerning the PBHM). As for meteorologic uncertainties, we specifically chose the hindcast-forecast architecture to address some uncertainties and characteristics of the meteorologic input data. First, the chosen architecture provides a more or less straightforward way of including meteorologic ensemble predictions in the future. This can be achieved by simply modifying the forecast LSTM, whilst no modification of the hindcast LSTM is required. The reason for this being that the hindcast data incorporates observations (ground stations and radar), whilst he forecasts heavily rely on numeric weather predictions. This leads to the fact that the forecast LSTM also has the potential to learn from uncertainties in the meteorologic forecasts. (**This was partly already included in the original manuscript, but was further highlighted**)

Unfortunately, for this study we did not have meteorologic forecasts available but addressing this topic is planned in the future. As for the hydrologic model, it is evident, that some underlying problems exist, given its poor performance. Noteworthy, this specific catchment was part of a broader study, where multiple catchments were modelled. Interestingly, most neighbouring catchments achieved quite a high model accuracy (NSE values slightly below and above 0.8) – calibration was done equally. In our opinion the poor performance of this catchment is a result of various uncertainties, one definitely being the inability of the employed HBV model to capture some important process in the catchment. (**This can also be found now in the discussion**)

Furthermore, changing environmental conditions can lead to a change of the outputs of the hydrologic model (which the ML model has not yet seen), i.e., the calibrated parameters are not well suited anymore to depict the catchment processes. In our opinion an effective, yet simple, way to reduce these uncertainties is regular retraining of the LSTM model. This may be done automatically, or manually. (**This is also present in the discussion, where we highlight the advantages of our cross-validation strategy**)

**AC Comment to RC1 – Initial response and planned activities**

We are thankful for the referee's valuable time in helping to improve the manuscript. This document will address all reviewer comments (gray boxes). We believe that the proposed changes will majorly improve the manuscript.

Considering the referee's comments, our planned improvements for the revised submission can be summarized as follows:

(1) In our opinion, the main critique of referee 1 concerned the models selected for comparison. We fully agree that comparing models that use additional exogenous variables, such as precipitation, with models that do not, results in an unfair comparison. To address this, referee 1 suggested to include a total of four models, namely: The original ARIMA and PBHM-HLSTM models as well as an ARIMA model that utilizes exogenous variables (ARIMAX) as well as an LSTM model that uses the same input variables as the original ARIMA model, termed eLSTM. We ran all models suggested by referee 1 and observed that the ARIMAX model did not achieve an improved forecast quality when compared to the already presented ARIMA model (see results below). For this reason, we see no added benefit of incorporating the ARIMAX model into the revised manuscript. Consequently, we suggest the following model selection for the revised manuscript:

a. **ARIMA**: This is the original ARIMA model presented in the preprint. This model utilizes simulated and observed runoff in the hindcast for correcting the hydrologic model's forecasts.

b. **eLSTM**: This model posses the proposed hindcast-forecast architecture of the PBHM-HLSTM but will only ingest simulated and observed runoff in the hindcast and simulated runoff in the forecast (similar to ARIMA). This model can be seen as an intermediate between ARIMA and the final PBHM-HLSTM, which in our opinion results in a more fair comparison between the individual models, as it features the data used for ARIMA but the architecture of the PBHM-HLSTM.

c. **PBHM-HLSTM**: This is the original PBHM-HLSTM model of the preprint.

(2) Another comment concerned the methodology of how we evaluated whether the LSTM is benefiting from the simulated runoff or not. We used overall statistics and the generalization capabilities as a proxy for judging if the LSTM benefited from the PBHM's results or not. Referee 1 suggested to perform this investigation by doing a sensitivity analysis instead (like integrated gradients or ingesting noise in the model), which in our opinion has the potential to majorly improve the manuscript. We already performed some analysis and will incorporate them in the revised manuscript as follows:

a. The LSTM (HLSTM) that does not include the simulated runoff will be excluded from the manuscript, as referee 1 suggested.

b. Instead, we will perform a sensitivity analysis using integrated gradients. We will present these results in terms of annual means and at the investigated flooding events (see preliminary results below)

(3) As referee 1 pointed out correctly, the flaw in the study design is also reflected in the formulated research questions. This will be changed accordingly to align with the major updates presented in points (1) and (2).

(4) Another comment addressed the choice of the hyperparameters of the LSTM, i.e., the model was too large. We fully agree that less hidden nodes should be used for regional studies. The models were already retrained and will be updated in the revised version of the manuscript (see below).

(5) We will also address all minor comments as stated below.

In our opinion, the above presented modifications address all major comments of referee 1 and their implementation will majorly benefit the final manuscript.

Dear editor and authors,

The following comment details my review of the manuscript "Long Short-Term Memory Networks for Real-time Flood Forecast Correction: A Case Study for an Underperforming Hydrologic Model" submitted to HESS.

In this preprint the authors present a model comparison study in which two (or three depending on the application) models are compared in their ability to forecast runoff. The models compared are all statistical- or machine learning-based models which take as inputs predictions of an underperforming conceptual model. The preprint is well written and the results are compelling. The scope of the manuscript is well suited for HESS and it has potential to be a great

contribution to the literature on runoff forecasting, as well as models which combine physics-based and data-driven approaches.

However, I have a number of major and minor comments/suggestions that should be addressed before final publication and ultimately will benefit the manuscript and overall study.

**Major Comments**

*The comparison is not "fair"*

What the ARIMA model is doing is very different than what the LSTM-based models are doing and this "unfair" comparison is apparent in the results. Evidently the model which is able to use data from precipitation in its forecasting step will be better at predicting events that have precipitation as its main driver and not the current or past discharge as calculated by an underperforming PBHM.

What is missing is a model that is in-between the ARIMA and HLSTM-PBHM and bridges the gap between the two approaches. In principle this could be an ARIMA which considers exogenous inputs (ARIMAX) or an LSTM which predicts errors without the aid of external variables for a direct comparison with the presented ARIMA model. This way we see how performance changes from having a model that is only correcting the PBHM (ARIMA), to a model which relies in the PBHM but can use other inputs when the PBHM fails (ARIMAX), to a model that accounts for all available input data and chooses what to use (HLSTM-PBHM).

Ultimately, I think the models which should be part of the study are: ARIMA, ARIMAX, LSTM which predicts errors only using $Q_{sim}$ and $Q_{obs}$ (name: $_e$LSTM?), and the presented HLSTM-PBHM.

We agree with the referee's comment that a direct comparison between a model that utilizes exogenous variables (LSTM), especially precipitation, and one that does not (ARIMA), is unfair. The original aim of this study was to compare an approach that can be regarded as more classic (ARIMA), as it is implemented in many existing forecasting systems, with a probably more capable LSTM model. However, we agree that the manuscript needs a revision to allow for a more fair and hence more objective comparison between both models presented. To achieve this, we tested the models suggested by referee 1, namely: ARIMA, ARIMAX (ARIMA with exogenous variables), eLSTM and the proposed HLSTM-PBHM. For the ARIMAX model, we included the maximum precipitation as an exogenous variable as this is the major driver for flooding in this catchment. We tried multiple model configurations, i.e., using the precipitation in the hindcast and also in the forecast by shifting the precipitation time series. However, it showed that independent of the configuration used, the exogenous variable did not have a positive impact on the forecast quality (see plots below). For this reason, we do not see an added benefit of including the ARIMAX model into the revised manuscript.

Considering this, we suggest comparing the three remaining models suggested by referee 1, namely: ARIMA (identical to the model in the preprint), eLSTM (new model) and HLSTM-PBHM (the proposed model of the preprint). Thereby the eLSTM model can be interpreted as an intermediate model lying between ARIMA and the HLSTM-PBHM. The reasoning for this is that the eLSTM uses the same data as ARIMA (more or less fair comparison between ARIMA and eLSTM) and the same architecture as the HLSTM-PBHM (more or less fair comparison between eLSTM and HLSTM-PBHM).

As for the suggested eLSTM, only ingesting Qsim and Qobs, the results show a significantly better performance compared to ARIMA, but also worse performance compared to the HLSTM-PBHM model, fed also with precipitation data, which aligns with our expectations. In our opinion this will be interesting to add to the revised manuscript and will improve the comparability between the model variants.

[Figure]

Furthermore, although the HLSTM was added to address a specific concern regarding the combination of the PBHM and an LSTM, I don't think its able to address this issue effectively as the authors also recognize by saying that in their findings: "We did not find strong evidence of whether the inclusion of the PBHM's results benefited the accuracy of the LSTM." My suggestion is that the HLSTM is completely dropped. Give that this model simply serves to check if $Q_{sim}$ is somewhat informative to the LSTM in the HLSTM-PBHM, this could be made clearer through a sensitivity analysis and not using a different model with also a different architecture. I suggest the sensitivity analysis could be done using integrated gradients as Kratzert et al. (2019) or simply by replacing the input of $Q_{sim}$ for noise and seeing the effect it has on the predictions by the model. If the LSTM does not consider the input from $Q_{sim}$ useful, there should be no effect by replacing this input for noise and vice versa.

We fully agree with referee 1 that our original methodology (evaluating overall statistics and the generalization capabilities) was not well suited to demonstrate the impact of the PBHM's results on the proposed LSTM model. We welcome the referee's suggestion to perform a sensitivity analysis instead. We tested both suggested methodologies, replacing the features with noise and also the integrated gradients method. In our opinion the most meaningful insights were gained when applying integrated gradients as follows:

The Integrated Gradient Method is used to evaluate the importance of an input of interest for the evaluated model's output and is defined as follows:

$$IntegratedGrads_i^{approx}(x) := (x_i - x_i') \times \sum_{k=1}^{m} \frac{\partial F\left(x' + \frac{k}{m}(x - x')\right)}{\partial x_i} \times \frac{1}{m}$$

where $x$ is the input of interest, $F(\circ)$ is the model, $x'$ is the baseline (in our case a sequence of zeros as suggested by Kratzert, 2019), $x_i$ is the input in the $i^{th}$ dimension, i.e. at the $i^{th}$ input node, and $m$ is the step size of the approximation of the integral (here 200, being within the suggested range by Sundararajan, 2017).

In our case, the output of the model is a sequence of size 96, representing the forecasting steps. The number of input dimensions, i.e. input nodes, accumulates from five hindcast and four forecast features, each sequences of size 96, to a total of 864 integrated gradients per output node and sample. To analyze this vast amount of information, we decided on two processing approaches to evaluate (i) **the importance of the nine features for a whole year (in the presented case 2017).** Noteworthy, in the revised manuscript this will be evaluated for all folds, and (ii) **their importance at a flood peak events**, as this is the most relevant in flood forecasting.

(i)    The whole year 2017 consists of 34903 samples. First, the integrated gradients for each sample are derived from the sum of the output sequence in respect to all 9x96 input nodes, i.e. dimensions. Then, the absolute values of these gradients are averaged over the whole year leading to 9x96 values, representing the input node importance for the whole output sequence.

A graphical representation of these results is shown in the following figure:

[Figure]

The left side of the figure shows the importance for each hindcast input node for the whole output sequence. It comes clear, that the input nodes at the end of the sequence (being closer to the forecast) have a much higher importance as the once at the beginning of it. Also, the precipitation feature generally shows very low importance, which can be explained due to the precipitation being zero throughout most of the year.

On the right side, the importance of each forecast input node is displayed. Generally, the precipitation features (*Pmean* and *Pmax*) show less impact on the total model output compared to simulation and temperature features (*Qsim* and *Tmean*).

The table below shows the sum of the input node importance for each feature, i.e. feature importance, which equals to the gray area under the curve as displayed in the figure above. Comparing the values for hindcast and forecast features individually – as these are basically two separate networks stacked on one another – the simulation *Qsim* is the most important forecast feature followed by the mean temperature *Tmean*. Regarding the hindcast, the measured runoff *Qmeasval* is the most important feature, followed by the temperature *Tmean*.

|  | Qsim | Pmax | Tmean | Pmean | Qmeasval |
|---|---|---|---|---|---|
| hindcast | 5.94 | 0.45 | 9.58 | 0.85 | **16.40** |
| forecast | **24.60** | 4.34 | 21.44 | 10.76 | |

(ii)   The feature importance for a flood peak event is calculated as the sum of the node importance over 96 consecutive samples. These samples range from the first time the flood peak appears in the forecasting horizon, namely at lead time step 96, until it becomes the next-step-forecast at lead time step 1. The integrated gradients are derived for each sample from the output's maximum. This ensures that the evaluations always take place at the output's peak.

The figure below shows the corresponding node importance for the first sample when the peak of the output is located at lead time step 96. It can be seen, that the importance of all the input nodes of the forecast is highest right before the output peak. Thereby, the precipitation features *Pmean* and *Pmax* show the highest importance.

[Figure]

The next plot displays the node importance when the output peak is located at lead time step 60 (sample 36). Again, the importance of all the input nodes of the forecast is highest right before the output peak, while precipitation features' importance is the highest overall.

[Figure]

To summarize these findings, we calculated the sum of the node importance per feature for each sample and visualized it according to the peak position in the forecast window. The figure below then shows the individual feature importance for a peak output at the corresponding lead time step.

[Figure]

It can be seen, that the precipitation forecast features (*Pmean* and *Pmax*) have the highest impact on the peak in the output sequence of the model, while the simulation *Qsim* has the lowest.

The conclusion of this is that, in general, the LSTM heavily relies on the PBHM's simulations when no precipitation/flooding events are present. However, the PBHM's results loose importance at peak flow predictions, possibly due to the poor initial guess of the underlying hydrologic model. Interestingly, also the temperature has a high importance considering the annual mean, which possible can be attributed to the model gaining some seasonality information from this feature. A more comprehensive analyses of these results will be presented in the revised manuscript.

Kratzert, Frederik, and Herrnegger, Mathew, and Klotz, Daniel, and Hochreiter, Sepp, and Klambauer, Günter. (2019). NeuralHydrology – Interpreting LSTMs in Hydrology. 10.1007/978-3-030-28954-6_19.

Sundararajan, Mukund, and Ankur, Taly, and Qiqi, Yan. (2017). Axiomatic Attribution for Deep Networks. Proceedings of Machine Learning Research. 70. https://arxiv.org/abs/1703.01365.

> This flaw in the design of the study is also reflected on the research questions established in the introduction. None of the research questions concern the ARIMA model, so why is it part of the study at all? From the point of view of the RQs, the study should only focus on the HLSTM-PBHM and the previously described *e*LSTM which could be considered a deep-learning adaptation of ARIMA while the HLSTM-PBHM is more akin to an ARIMAX, keeping the scope of the paper within error correcting strategies, and then the discussion can focus on the benefit of precipitation as an input during forecasting, and the difference between years where the PBHM is acceptable (2014 and 2016) in contrast to when it's terrible (2017).

We agree with the referee that the shortcomings in the study design are also reflected in the established research questions. Overall, we do believe that the research questions should clearly reflect the intentions of the study. The original idea of the study was to assess the potential of LSTM models to correct poor forecasts in cases where ARIMA models may result in unfavorable results. In our opinion this very relevant in the field of flood forecasting as many operational forecasting systems currently rely on ARIMA type correction strategies. So we do believe that the research questions should be changed accordingly to still reflect the intentions of the study, while adding the valuable suggestions of the referee:

(1) RQ1: Research question one will deal with an overall comparison of the ARIMA results with the proposed PBHM-HLSTM and the intermediate eLSTM model. ARIMA in this regard can be seen as the classic approach employed in many forecasting systems (We will back this up with more literature of existing forecasting systems).
(2) RQ2: Research question two will deal with a comparison of the peak runoff prediction performance, as this is the most important thing in operational flood forecasting
(3) RQ3: This research question will still deal with the usefulness of the PBHM on the LSTM forecasts but evaluated with the suggestions given by referee 1. In our opinion this is a very important point as the question "why not simply replace the PBHM with a LSTM?" is a very valid and crucial one given the poor predictive skills of the original PBHM.

> *Hyperparameters of the LSTM-based models*
>
> In the supplemental information of the paper by Nevo et al. (2022), their model is described to have an LSTM of 128 hidden units for hindcast and another 128 hidden units for forecast, which is similar to the 96 used in this article, but in the case of Nevo et al. (2022), the model was trained to forecast in at least 165 basins which use LSTM for the "stage forecast model". The architecture presented by Nevo et al. (2022) is also based on the MTS-LSTM presented by Gauch et al. (2021). Although the purpose of that second paper is not forecasting, the idea of "handing" the hidden states from one LSTM to another is the same, and in their case both LSTMs which send and receive the hidden states have 64 hidden nodes. This is also applied in a regional case study in which the amount of data that the model needs to ingest is a lot larger. Finally, in a more recent example by some of the same authors of the previous papers, Kratzert et al. (2024) train single-basin LSTMs using models with hidden nodes ranging from 8 to, at most, 32. This is not a criticism of not using LSTM in a regional setting, in my view LSTMs are

still valid for application in a single basin, acknowledging their limitations, but their size shouldn't be the same as those used in regional modelling.

This could be addressed by adding smaller sizes into the hyperparameter search space and I would encourage the authors to present their results for training/validation in supplemental material of the article in the form of loss curves, metrics in training/validation, etc.

We agree with referee 1 that the amount of nodes should scale with the complexity of the task at hand, i.e., a regional model should be less complex than for cases with multiple basins. We will address this issue by retraining (most of it has been done already as shown above) the models with a decreased search space using 4 to at most 32 units.

We agree with the referee that it is beneficial to also publish training/validation results (losses, metrics, etc.), which will be added to the revised manuscript.

Checking the code repository and additional files provided by the authors, the '\tb' folder was not included so the TensorBoard logs cannot be checked.

On checking the code further, there are remainder classes and functions that were not part of the study such as the models that include 'CNN' layers. I would suggest a general cleanup, but the code definitely runs and I was able to generate one of the trials done for the study and the corresponding logs.

We agree with the referee that we should have published all logs of the hyperparameter tuning from the beginning. We are going to include all TensorBoard logs together with a cleaned-up version of the code.

Taking a look at some of the post-processing notebooks, I find the information presented in the `notebook_aux_peak_events.ipynb` to be informative and would encourage the authors to include some of the plots for peak events in section 4.2 in the manuscript. The colors for the ARIMA model need to be adjusted though, because they are the same as the "measured" data.

We decided to remove these plots from the manuscript due to structural reasons but kept them in the code files as we also find them quite informative. For the final submission, we will try to include them in section 4.2 as suggested, or at least add them to the supplement material.

**Minor Comments**

General: I suggest that the name of the HLSTM-PBHM should be flipped around to PBHM-HLSTM as the PBHM is the initial step in the pipeline.

We agree and will change this accordingly in the revised manuscript.

Line 72: The hindcast-forecast LSTM cannot be called "novel" as it is adapted from the approach of Nevo, et al. (2022), which in-itself is adapted from other sources.

We agree and will change this accordingly.

Line 146: How was the search-space for the hyperparameters of the ARIMA model defined?

The search space for ARIMA was defined based on a prior testing. However, in the final version of the manuscript, we will retrain the model with an updated search space. The search space will be determined to also align with the comments of referee 2. Specifically, we will determine the p and q orders based on the PACF and ACF functions. As for the differentiation order (d), really

only d=1 produced reasonable results. So this will be also addressed as it makes no sense to still include lower (0) or higher orders to the search space if this does not produce nearly as good results. For more details also see our comments to referee 2.

Line 171: Why was a loss function which combines two metrics chosen? Was minimizing NSE or simply MSE tried? From Appendix B the authors say that this was adapted from Nevo et al. (2022) but in that paper they minimize a negative log-likelihood as they have a probabilistic model.

Unfortunately, the citation of Nevo et al. (2022) was misplaced in this context. We apologize for the inconveniences and will fix this in the revised submission.

The first versions of the models were trained using the MSE as a loss function. At some iteration, we switched to the presented loss functions.To us it made sense to use the same function for model training (loss function) and for the tuner objective. Nevertheless, throughout the rerun of our experiments on the smaller search space, we compared the differences between our employed loss function and a simple MSE more extensively. The results showed no added benefit of the employed loss function (presented in the manuscript) over a simple MSE. Interestingly, the MSE even produced slightly better results overall. The figure below shows the histogram of the scores of the 50 trials during the hyperparameter tuning. For both, the eLSTM and the PBHM-HLSTM, the model trained on the MSE shows a better score on average as well as at the best trial.

For the final submission, we will therefore choose the models trained using the MSE as the loss function.

[Figure]

Figure 4: On the right-hand side of the figure I'm missing the $Q_{forecast}$ as in Figure 3. Further the "Targets" should be dropped from this figure or added to Figure 3 to have somewhat equivalent descriptions of the model architectures.

We will do a rearrangement of the graphic to include Qforecast for the final submission. Furthermore, we will remove the targets from Figure 3, as suggested by the referee.

Table 2: Although PBIAS is a good overall metric to include, these results would greatly benefit of including metrics which directly target low- and high-flows like FHV and FLV. See Gauch et

al. (2021) or directly Yilmaz et al. (2008). Some of the results described in the text using PBIAS are more suited to be described using these two metrics.

We agree that the suggested metrics FHV and FLV are more suited to argue about performances for base flow and peak flow conditions, respectively. For this reason we will include the FHV and FLV metrics. We will further investigate if it makes sense for the revised manuscript to replace the PBIAS or if it still has a benefit for interpreting the presented results.

Also, I find the reported PBIAS metrics for the ARIMA models to be strange given their KGE and NSE per year. From the code, in the `run_arima.py` I see it calls the `calculate_bias` function but only prints values to the screen while in `notebook_tab1-4_table_results.ipynb` the metrics are read directly from a file. I'm guessing that `metrics.txt` is generated using the data in the other files in that folder `metric_nse.txt`, etc. but I also don't see where in the code those files are dumped. `run_arima.py` dumps the `all_fc_df` as a pickle, but I'm not sure if that DataFrame is used to generate the metrics `.txt` files.

Yes, we were also surprised that the BIAS error of the ARIMA model was that low. However, upon further investigation, it showed that ARIMA produced many "perfect" forecasts throughout the year. This was especially true in baseflow conditions, which had a large positive impact on the PBIAS values. Noteworthy, this is also the reason why ARIMA in some instances has quite good KGE – KGE has a direct term for the BIAS - statistics, whilst the NSE is not that good. Upon inspecting the results using the FHV and FLV metrics, ARIMA's forecasts came out as less favorable. So we agree with the referee, that the PBIAS might not be a good metric in this case to draw conclusions from, which will be addressed in the revised manuscript.

The referee is right. For the ARIMA model, the metrics are computed from the 'all_fc_df'. We will change this accordingly such that the metrics are computed directly, which majorly improves the reproducibility of the results.

Table 3: Generalization refers to a models ability to predict in previously unseen data drawn from the same distribution as the one used to train the model. In doesn't have to do with differences between validation and testing. I would consider correcting this table with differences between the testing and training sets, or not including it at all as most of the discussion cantered around these results appears or can be included in other sections.

We agree with the referee that the term generalization was misused in the original version of the manuscript. However, based on the referee's prior suggestion of including a sensitivity analysis for evaluating the benefit of Qsim on the model results, this evaluation becomes obsolete and will be removed from the revised manuscript.

Fig. 6 and Fig. 7: The legend of both columns should not be shared. Currently it appears as if the "normalized win ratio" has something to do with the standard deviation of each model.

We agree with the referee and we will change the plots accordingly to avoid confusion.

**References**
Gauch, M., Kratzert, F., Klotz, D., Nearing, G., Lin, J., & Hochreiter, S. (2021). Rainfall–runoff prediction at multiple timescales with a single Long Short-Term Memory network. Hydrology and Earth System Sciences, 25(4), 2045–2062. https://doi.org/10.5194/hess-25-2045-2021

Kratzert, F., Gauch, M., Klotz, D., & Nearing, G. (2024). HESS Opinions: Never train an LSTM on a single basin. Hydrology and Earth System Sciences Discussions, 1–19. https://doi.org/10.5194/hess-2023-275

Kratzert, F., Herrnegger, M., Klotz, D., Hochreiter, S., & Klambauer, G. (2019). NeuralHydrology—Interpreting LSTMs in Hydrology. arXiv:1903.07903 [Physics, Stat], 11700, 347–362. https://doi.org/10.1007/978-3-030-28954-6_19

Nevo, S., Morin, E., Gerzi Rosenthal, A., Metzger, A., Barshai, C., Weitzner, D., Voloshin, D., Kratzert, F., Elidan, G., Dror, G., Begelman, G., Nearing, G., Shalev, G., Noga, H., Shavitt, I., Yuklea, L., Royz, M., Giladi, N., Peled Levi, N., … Matias, Y. (2022). Flood forecasting with machine learning models in an operational framework. Hydrology and Earth System Sciences, 26(15), 4013–4032. https://doi.org/10.5194/hess-26-4013-2022

Yilmaz, K. K., Gupta, H. V., & Wagener, T. (2008). A process-based diagnostic approach to model evaluation: Application to the NWS distributed hydrologic model. Water Resources Research, 44(9). https://doi.org/10.1029/2007WR006716

**AC Comment to RC2 – Initial response and planned activities**

We are thankful for the referee's valuable time in helping to improve the manuscript. This document will address all reviewer comments (gray boxes). We believe that the proposed changes will majorly improve the manuscript.

Considering the referee's comments, our planned improvements for the revised submission can be summarized as follows:

(1) We will conduct and present a more in-depth statistical analyses of the residuals, as also shown in the Literature suggested by the referee. As the referee pointed out, many studies report a high degree of heteroscedasticity in the residuals. Our analysis indicate the same (see evaluations below). To address this, we will apply a Box-Cox transformation to the data to decrease the degree of heteroscedasticity. Furthermore, we will investigate and report the ACF and PACF and also use this information for redefining the search space of our ARIMA model (see also the comment from referee 1).

(2) We agree with the referee that the HLSTM-PBHM does not perform a direct error correction as not the residuals are forecasted but the discharge directly. We will address this accordingly in the relevant sections of the manuscript (especially the title).

In our opinion, the above presented modifications address all major comments of referee 2 and will majorly benefit the manuscript.

This study proposes an application of Long Short-Term Memory Networks (LSTM) complemented by the results of a hydrological model (PBHM) for operational flood forecasting in a smaller mountainous catchment. The performance of the resulting HLSTM-PBHM is compared with an ARIMA error correction model and a standalone application of the LSTM (HLSTM).

The results of this study are particularly significant, as they reveal performance improvements for the HLSTM-PBHM, especially for larger lead times. These findings have practical implications for flood forecasting in similar catchments.

The paper is within the scope and very interesting for the readers of HESS. The authors address a topic of high relevance for flood forecasting since studies focusing on small catchments and requiring sub-daily time steps are limited.

> The authors have done a commendable job of presenting the scientific results concisely and well-structured. However, I have some fundamental comments on the interpretation of the proposed method and the concept of the experimental design to compare the different approaches:

> From my perspective, the proposed HLSTM-PBHM is an informed approach that uses precalculated results of the hydrological model (PBHM) combined with observations for the hindcast rather than applying an explicit error correction as the ARIMA error correction model does. Therefore, the title of the paper should reflect this, and I suggest revising it.

We agree with the referee's comment that, in principle, the ARIMA and LSTM models follow a different mythology of how the final forecasts are obtained. ARIMA forecasts the residuals, which are used to correct the simulations of the hydrologic model, while the LSTM uses the hydrologic modelling results as a feature to directly forecast the runoff. This implies that the LSTM does not apply an error correction, which should be revised in the relevant sections of the manuscript, especially the title. We will thus change the title and also make this fact more clear in other sections of the manuscript.

> This approach's consequence is that the input data used and the internal corrections of HLSTM-PBHM cannot be compared with the residual errors of the hydrological model and the corrections calculated by the explicit error correction models.

It has to be mentioned that we previously tried a residual LSTM model. However, the prediction accuracy was inferior compared to the presented LSTM variant that directly predicted the streamflow. The reason for this might be that the predictions of the hydrologic model (in this study) are very poor when for instance compared to the study cited by the referee, where the original model's performance was already quite good. This leads to the fact that the LSTM does not rely that much on the simulated stream flow in some conditions (i.e., at peak runoff). This was demonstrated in our reply to RC1 and will be included in the final manuscript.

A general shortcoming of comparability between ARIMA and the HLSTM-PBHM was also pointed out by referee 1. We will address this issue by introducing an intermediate model that uses the same data as ARIMA and has the same architecture as the HLSTM-PBHM. This, however, still does not resolve the fact that residual errors across the models cannot be compared like it was done in the literature suggested by the referee. One option would be to compute the LSTM's residuals **after** the streamflow was forecasted. However, we do believe that a more objective way (for this specific study!) is still comparing the final forecasts of each model. However, we will make the fact that one model forecasts residuals and one runoff directly more clear throughout the manuscript.

> In general, a comprehensive analysis of the residual errors of the PBMH model, e.g., the underlying statistical distribution, would be helpful and give the reader more insight to interpret the results. It would also prove the assumption of whether the errors are normally distributed. Many studies (among them [1]) found a high heteroscedasticity variance of residuals, which should be checked and considered for the residuals in the study.

We will add the residual statistics of the PBHM to address the comment of referee 2. Similar to other studies, the literature pointed out by the referee amongst them, we also found a high degree of heteroscedasticity in the residuals. We will reduce the degree of heteroscedasticity by applying

a Box-Cox transformation as suggested in the literature presented by the referee (see figure below).

[Figure]

We will also report on the autocorrelation functions (PACF and ACF) and use this information for redefining the search space for ARIMA but also to discuss the implications on the LSTM as shown in the literature suggested by the referee.

Please also briefly introduce the PBHM model in the Methods Section as it is used in the study.

We will include more information on the PBHM in the revised manuscript such that the reader gets a general picture of the model without having to go to the cited literature.

Multiple errors exist in flood forecasting due to meteorological uncertainties and those rising from the structure and parametrization of the hydrological model. Please elaborate on how the different contributions could be considered in future developments of HLSTM-PBHM in the discussion.

We agree with the referee that a variety of uncertainties exist in operational flood forecasting (e.g., meteorology, streamflow observations and all uncertainties concerning the PBHM). As for meteorologic uncertainties, we specifically chose the hindcast-forecast architecture to address some uncertainties and characteristics of the meteorologic input data. First, the chosen architecture provides a more or less straightforward way of including meteorologic ensemble predictions in the future. This can be achieved by simply modifying the forecast LSTM, whilst no modification of the hindcast LSTM is required. Second, the chosen architecture allows for a clear distinction between meteorologic hindcast and forecast data. This is especially relevant, since often quite large differences between meteorologic hindcast data and forecast data are present. The reason for this being that the hindcast data incorporates observations (ground stations and radar), whilst he forecasts heavily rely on numeric weather predictions. This leads to the fact that the forecast LSTM also has the potential to learn from uncertainties in the meteorologic forecasts. Unfortunately, for this study we did not have meteorologic forecasts available but addressing this topic is planned in the future. As for the hydrologic model, it is evident, that some underlying problems exist, given its poor performance. Noteworthy, this specific catchment was part of a broader study, where multiple catchments were modelled. Interestingly, most neighbouring catchments achieved quite a high model accuracy (NSE values slightly below and above 0.8) – calibration was done equally. In our opinion the poor performance of this catchment is a result of

various uncertainties, one definitely being the inability of the employed HBV model to capture some important process in the catchment.

Furthermore, changing environmental conditions can lead to a change of the outputs of the hydrologic model (which the ML model has not yet seen), i.e., the calibrated parameters are not well suited anymore to depict the catchment processes. In our opinion an effective, yet simple, way to reduce these uncertainties is regular retraining of the LSTM model. This may be done automatically, or manually.

---

## Referee Report (RR1)

**Review of**

**"Long Short-Term Memory Networks for Enhancing Real-time Flood Forecasts: A Case Study for an Underperforming Hydrologic Model"- revision 1**

by Sebastian Gegenleithner, Manuel Pirker, Clemens Dorfmann, Roman Kern, and Josef Schneider

**General comments:**

The authors considered the reviewer's comments, addressed the concerns, and considerably improved the quality of the paper. The revisions are acceptable. The experimental design and description changes significantly improve coherency and clarity. The paper is now acceptable for publication. I congratulate the authors for this nice piece of scientific work!

**Detailed comments:**

- timing error and peak error should already introduced in the "Methods Section 3.2". The symbol for the timing error is somewhat misleading: maybe $e_{\Delta t}$ would be more appropriate.

---

## Author Response (AR2)

**Author's comments to referees**

Again, we express our gratitude to the referees and the handling editor for the time they invested in helping to improve our manuscript. As before, we agree with all comments and believe that the implemented changes further enhance the quality of the paper. Below you find our detailed response to the referee comments:

**AC Comment to RC1 – Implemented changes**

> Code: I reviewed the code on the Zenodo repository and was able to  follow the README to check that both scripts and notebooks worked  correctly. The scripts (mainly the tuner) I didn't run until termination  as it was not my intention to recreate any models, only checking if  they were working correctly. My only suggestion would be to add a  ´requirements.txt´ file to the Github repo containing the same  information as the 'environment.yml' as not everyone uses Anaconda as  their package manager (I don't). I think it's worth the trouble because  the Github repo can be a very good starting point for someone just  getting started in a similar application of ARIMA models and/or LSTMs.

A requirements.txt was added to the Github repo as well as to Zenodo

> Section 4.1: I would be careful in using words such as "exceptional" or  "outstanding" to describe the evaluation results of a particular model  since performance assessment should rely on quantitative metrics rather  than qualitative judgment. Also, there are cases in which there is no "winner" depending on the metric and/or year analysed, therefore the use  of these words feels overly positive and overstate the authors'  findings.

We agree with the referee on this point. We rephrased all relevant sections that sounded overly positive and excluded wordings like "exceptional" or "outstanding".

> ARIMAX: In the current manuscript, a direct comparison between the ARIMA  and eLSTM models can be made, but I feel there's still a gap due to the  absence of a comparable benchmark for the PBHM-LSTM model. An ARIMAX  model could fill this gap, though I understand its exclusion aligns with  the original research focus on improving upon the commonly used ARIMA  model. That said, the authors' response shows their test of an ARIMAX model in which they finding no significant differences compared to the  ARIMA model included in the paper. I believe that including this detail  in the manuscript would be beneficial as it adds context.

We now mentioned that an ARIMAX model was tested but did not yield significant differences to the presented ARIMA model in the discussion section. We also emphasized on the fact that this was true for this particular study, but should not be seen as a general statement.

**AC Comment to RC2 – Implemented changes**

> timing error and peak error should already introduced in the "Methods Section 3.2". The symbol for the timing error is somewhat misleading: maybe e$\Delta$t would be more appropriate.

The peak magnitude and timing errors are now introduced in the methods section 3.2. We also agree that the symbol of the timing error could be misleading. We changed this accordingly in the revised manuscript.